# Bayesian Ego-graph Inference for Networked Multi-Agent Reinforcement Learning

**Wei Duan**
Australian Artificial Intelligence Institute
University of Technology Sydney
Sydney, Australia
`wei.duan@student.uts.edu.au`

**Jie Lu**
Australian Artificial Intelligence Institute
University of Technology Sydney
Sydney, Australia
`jie.lu@uts.edu.au`

**Junyu Xuan**
Australian Artificial Intelligence Institute
University of Technology Sydney
Sydney, Australia
`junyu.xuan@uts.edu.au`

## Abstract

In networked multi-agent reinforcement learning (Networked-MARL), decentralized agents must act autonomously under local observability and constrained communication over fixed physical graphs. Existing methods often assume static neighborhoods, limiting adaptability to dynamic or heterogeneous environments. While centralized frameworks can learn dynamic graphs, their reliance on global state access and centralized infrastructure is impractical in real-world decentralized systems. We propose a stochastic graph-based policy for Networked-MARL, where each agent conditions its decision on a sampled subgraph over its local physical neighborhood. Building on this formulation, we introduce **BayesG**, a decentralized actor–critic framework that learns sparse, context-aware interaction structures via Bayesian variational inference. Each agent operates over an ego-graph and samples a latent communication mask to guide message passing and policy computation. The variational distribution is trained end-to-end alongside the policy using an evidence lower bound (ELBO) objective, enabling agents to jointly learn both interaction topology and decision-making strategies. BayesG outperforms strong MARL baselines on large-scale traffic control tasks with up to 167 agents, demonstrating superior scalability, efficiency, and performance.

## 1 Introduction

Multi-agent reinforcement learning (MARL) has emerged as a powerful framework for sequential decision-making in distributed systems, enabling applications in autonomous driving [1, 2], wireless communication [3, 4], multiplayer games [5, 6], and urban traffic control [7, 8]. A popular training paradigm is centralized training with decentralized execution (CTDE), where a centralized critic leverages global state information to train decentralized [9–12].

While CTDE methods demonstrate strong empirical performance in simulation benchmarks [13–15], they often rely on access to global observations and centralized learning infrastructure, as shown in Figure 1(a). This assumption rarely holds in real-world applications, where agents are geographically distributed and subject to local sensing and communication constraints. Instead, many practical domains—from urban mobility to smart grids—are better modeled as *networked MARL* [16–19],

where agents interact over a fixed communication graph and can only observe or exchange information with nearby neighbors.

A major challenge in networked MARL is the use of static communication graphs, where agents are hardwired to exchange information with all local neighbours regardless of contextual relevance [16, 17]. This can lead to inefficient coordination, unnecessary message exchange, or even performance degradation, particularly in dynamic settings like traffic control, where congestion varies over time and not all neighbours are equally informative. This raises a fundamental question:

> *Can **decentralized agents** learn to **dynamically adapt their interaction structure** using local observations and task feedback?*

While recent CTDE methods have explored learning dynamic interaction graphs [20, 15, 21, 22], their applicability is limited in decentralized settings, where agents must reason over local observability and adhere to the physical structure of the environment.

To address this, we propose **BayesG**, a decentralized graph-based actor–critic framework for networked MARL that learns latent interaction structures via Bayesian inference. As illustrated in Figure 1(b), we begin by introducing a *graph-based policy*, where each agent conditions its decisions on a stochastic subgraph sampled from a learned distribution over its physical neighbourhood. Each agent operates over an *ego-graph*, a localized subgraph capturing its immediate neighborhood, enabling context-aware decisions under topological constraints. We formulate this process as *Bayesian variational inference of latent graphs*, where each agent $i$ infers a binary mask $Z_i$ over its local neighbourhood. The mask is treated as a latent variable with posterior $p(Z_i \mid G_{\mathcal{V}_i}^{\text{env}}, D_i)$, conditioned on the physical subgraph $G_{\mathcal{V}_i}^{\text{env}}$ and agent-specific data $D_i$ (e.g., neighbour states, trajectories and polices). The posterior is approximated by a variational distribution $q(Z_i; \phi_i)$, optimized end-to-end via the evidence lower bound (ELBO).

BayesG integrates latent graph inference into policy learning, enabling agents to prioritize critical communication links within their local ego-graphs and prune irrelevant ones—all without requiring global supervision. This leads to task-adaptive, uncertainty-aware, and communication-efficient coordination under topological constraints. Experiments on both synthetic and real-world traffic control benchmarks show that BayesG outperforms state-of-the-art MARL baselines in both performance and interpretability.

**Our main contributions are:**

- We propose a stochastic graph-based policy for networked MARL, where each agent conditions decisions on a sampled subgraph over its physical neighbourhood.
- We formulate latent graph learning as Bayesian variational inference, treating edge masks as posterior distributions constrained by the environment topology and agent-local data.
- We develop an end-to-end training algorithm that integrates variational graph inference with actor–critic learning via an ELBO objective.

## 2   Related work

**Networked MARL.** Networked MARL focuses on decentralized learning over fixed topologies, often without explicitly adapting the communication structure. Consensus-based methods [16, 17] synchronize local value functions via neighborhood averaging, typically assuming partial or global observability. Other works incorporate network-aware priors such as distance-based decay [23] or local reward aggregation [19] to promote spatially coherent coordination. Model-based frameworks [24, 25] exchange predicted trajectories but still rely on static interaction graphs. Communication-based methods like NeurComm [7] enable information sharing across neighbors, yet use fixed topologies throughout training. Recent work [26, 23, 27] has explored sampling strategies to improve scalability in networked systems, but these methods sample from predefined or uniform distributions at the environment or agent level. Our method differs fundamentally by introducing a Bayesian latent graph inference mechanism that allows each agent to adaptively select task-relevant neighbors from its physical graph, enabling communication-efficient, locally grounded coordination.

**Graph-based MARL and Latent Structure Learning.** Recent advances in cooperative MARL leverage latent interaction graphs to enhance coordination and scalability. Methods such as DGN [28],

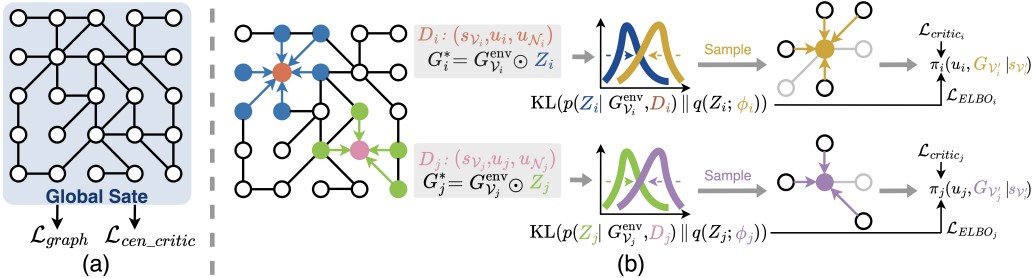

Figure 1: **(a)** In CTDE, the global state is available for learning both the centralized critic and the interaction graph. **(b)** Overview of BayesG. In networked MARL, each agent's state and action are influenced by its neighbors, forming local data $D_i = \{s_{\mathcal{V}_i}, u_i, u_{\mathcal{N}_i}\}$. We formulate latent graph learning as *Bayesian variational inference*, where each agent infers a binary mask $Z_i$ over its neighborhood from the environment graph $G_{\mathcal{V}_i}^{\text{env}}$ and local data $D_i$. The posterior $p(Z_i \mid G_{\mathcal{V}_i}^{\text{env}}, D_i)$ is approximated by a variational distribution $q(Z_i; \phi_i)$, from which a sparse subgraph is sampled and used for graph-conditioned policy learning via an ELBO objective.

DICG [29], and HGAP [30] employ attention mechanisms or graph neural networks (GNNs) [31, 32] to infer inter-agent dependencies and guide information exchange. Variational approaches [20, 22] further treat interaction graphs as latent variables, jointly optimizing communication structure and policy learning. However, most existing methods, including Dec-POMDP-based coordination graph approaches [33, 29, 34, 14, 21], assume centralized training with access to global state and reward signals, and allow unrestricted graph rewiring, which are impractical in real-world decentralized settings. In contrast, we address the more realistic *networked MARL* scenario by learning dynamic latent interaction masks on the environment graph using only local observations.

## 3 Preliminaries

We consider *networked MARL*, over a fixed interaction graph $G = (\mathcal{V}, \mathcal{E})$, where each agent $i \in \mathcal{V}$ interacts locally with neighbors $\mathcal{N}_i = \{j \mid (i, j) \in \mathcal{E}\}$ and defines a closed neighborhood $\mathcal{V}_i := \mathcal{N}_i \cup \{i\}$. At each step, agent $i$ receives a local observation $\tilde{s}_i = f(s_{\mathcal{V}_i})$, which may include its own state and aggregated information from neighbors, and selects an action $u_i \in \mathcal{U}^i$ according to a local policy $\pi_i(u_i \mid \tilde{s}_i)$. The encoder $f(\cdot)$ captures neighbourhood-level context using neural architectures (e.g., MLPs [7]); specific choices are detailed in Section 5.1.2. We begin with the spatiotemporal MDP formulation, followed by the decentralized actor–critic (A2C) training objective.

**Definition 1** (**Spatiotemporal MDP**[7]). *A spatiotemporal MDP is defined as the tuple $(\mathcal{S}, \{\mathcal{U}^i\}_{i \in \mathcal{V}}, \mathcal{P}, \mathcal{R}, \gamma, \zeta)$, where $\mathcal{S}$ is the global state space, $\mathcal{U}^i$ the action space of agent $i$, and $\zeta$ the initial state distribution. The local transition model of agent $i$ is:*

$$p_i(s_i' \mid s_{\mathcal{V}_i}, u_i) = \sum_{u_{\mathcal{N}_i}} \left( \prod_{j \in \mathcal{N}_i} \pi_j(u_j \mid f(s_{\mathcal{V}_j})) \right) p(s_i' \mid s_{\mathcal{V}_i}, u_i, u_{\mathcal{N}_i}), \quad (1)$$

*where the transition depends on local neighborhood states and actions. The local reward is $\mathcal{R}(s_{\mathcal{V}_i}, u_{\mathcal{V}_i}) \in \mathbb{R}$, and the objective is to maximize expected return: $\mathbb{E}_\pi \left[ \sum_{t=0}^\infty \gamma^t \mathcal{R}(s_{\mathcal{V}_i, t}, u_{\mathcal{V}_i, t}) \right]$, with discount factor $\gamma \in (0, 1]$.*

**Definition 2** (**Networked MARL with decentralized A2C**[7]). *Let $\{\pi_{\theta_i}\}_{i \in \mathcal{V}}$ and $\{V_{\omega_i}\}_{i \in \mathcal{V}}$ be decentralized actor and critic networks, respectively. Given an on-policy minibatch $\mathcal{B} = \{(s_{i,\tau}, u_{i,\tau}, r_{i,\tau}, u_{\mathcal{N}_i, \tau})\}$, the losses are:*

$$\mathcal{L}(\theta_i) = \frac{1}{|\mathcal{B}|} \sum_{\tau \in \mathcal{B}} \left[ -\log \pi_{\theta_i}(u_{i,\tau} \mid \tilde{s}_{i,\tau}) \hat{A}_{i,\tau}^\pi + \beta \sum_{u_i \in \mathcal{U}^i} \pi_{\theta_i}(u_i \mid \tilde{s}_{i,\tau}) \log \pi_{\theta_i}(u_i \mid \tilde{s}_{i,\tau}) \right], \quad (2)$$

$$\mathcal{L}(\omega_i) = \frac{1}{|\mathcal{B}|} \sum_{\tau \in \mathcal{B}} \left( \hat{R}_{i,\tau}^\pi - V_{\omega_i}(\tilde{s}_{i,\tau}, u_{\mathcal{N}_i, \tau}) \right)^2, \quad (3)$$

where $\tilde{s}_{i,\tau} = f(s_{\mathcal{V}_i,\tau})$ is the encoded local observation. The $\hat{A}^\pi_{i,\tau} = \hat{R}^\pi_{i,\tau} - v_{i,\tau}$ is advantage estimate, where the reward is $\hat{R}^\pi_{i,\tau} = \sum_{\kappa=0}^{K-1} \gamma^\kappa \sum_{j\in\mathcal{V}_i} \alpha^{d_{ij}} r_{j,\tau+\kappa} + \gamma^K v_{i,\tau+K}$,, and $K$ denotes the rollout horizon. The $v_{i,\tau} = V_{\omega'_i}(\tilde{s}_{i,\tau}, u_{\mathcal{N}_i,\tau})$ is the target critic output. The $\alpha \in (0,1]$ adjusts influence from distant neighbors, and $\beta$ controls the entropy regularization.

## 4 Method

We propose **BayesG**, a decentralized actor–critic framework for networked MARL that learns sparse, context-aware interaction graphs via variational inference. We begin by formulating a *graph-based policy*, where each agent conditions its decision on a sampled subgraph over its physical neighborhood. This formulation enables agents to adaptively modulate their coordination structure based on local observations while respecting communication constraints. We then extend this framework to optimize both the policy and graph sampling distribution jointly using a variational learning objective.

### 4.1 Graph-based Policy with Latent Interaction Structures

In conventional A2C for networked MARL (Definition 2), agents condition their policy on local observations $\tilde{s}_{i,\tau} = f(s_{\mathcal{V}_i,\tau})$, which implicitly encode neighbor influence but treat all neighbors as equally informative. This uniform treatment limits the agent's ability to adapt to dynamic environments or prioritize critical interactions. To overcome this, we introduce a *graph-based policy* in which each agent samples a subgraph from a learned distribution and conditions its decision on the sampled structure. (See Appendix A.1 for details).

**Definition 3** (**Graph-based Policy**). *The policy of agent $i$ is defined as:*

$$\pi_i(u_i, G_{\mathcal{V}_i} \mid s_{\mathcal{V}_i}; \theta_i, \varphi_i) = \rho(G_{\mathcal{V}_i} \mid s_{\mathcal{V}_i}; \varphi_i) \cdot \tilde{\pi}_i(u_i \mid \tilde{f}_i(s_{\mathcal{V}_i}, G_{\mathcal{V}_i}); \theta_i), \tag{4}$$

*where $G_{\mathcal{V}_i} \in \{0,1\}^{|\mathcal{V}_i|\times|\mathcal{V}_i|}$ is a sampled binary adjacency matrix drawn from distribution $\rho(\cdot; \varphi_i)$ over agent $i$'s closed neighborhood. The graph is restricted to the environment topology: $G_{\mathcal{V}_i} \in \mathcal{G}^{env}_{\mathcal{V}_i} := \{G \in \{0,1\}^{|\mathcal{V}_i|\times|\mathcal{V}_i|} \mid G \preceq G^{env}_{\mathcal{V}_i}\}$, ensuring only physically permitted connections. The function $\tilde{f}_i(\cdot)$ is a graph-conditioned encoder, and $\tilde{\pi}_i(\cdot \mid \cdot; \theta_i)$ is the action-selection policy.*

This formulation enables agents to stochastically select contextually relevant neighbors, driven by local observations. **By formulating coordination as a distribution $\rho(\cdot)$ over subgraphs, this approach provides a general framework for inferring context-dependent interaction structures**—allowing agents to explore diverse coordination patterns and adapt to non-stationary dynamics. In traffic signal control, this allows dynamically emphasizing congested junctions while suppressing irrelevant neighbors.

We now extend the A2C training objective to accommodate graph-conditioned policies. This graph-based loss serves as the actor component of our training framework and jointly optimizes the policy and interaction structure. (See Appendix A.2 for details).

**Definition 4** (**Graph-based A2C Objective**).

$$\mathcal{L}_{\theta,\varphi} = \frac{1}{|\mathcal{B}|} \sum_{\tau\in\mathcal{B}} \mathbb{E}_{G_{\mathcal{V}_i}\sim\rho} \left[ -\log \tilde{\pi}_i(a_{i,\tau} \mid \tilde{f}_i(s_{\mathcal{V}_i}, G_{\mathcal{V}_i})) \cdot \hat{A}^\pi_{i,\tau} + \beta \cdot \mathcal{H}(\tilde{\pi}_i(\cdot \mid \tilde{f}_i)) \right], \tag{5}$$

*where the entropy term is defined as $\mathcal{H}(\tilde{\pi}_i(\cdot \mid \tilde{f}_i)) = -\sum_{u_i\in\mathcal{U}^i} \tilde{\pi}_i(u_i \mid \tilde{f}_i)\log\tilde{\pi}_i(u_i \mid \tilde{f}_i)$, and $\mathcal{B}$ denotes an on-policy trajectory batch, and $\hat{A}^\pi_{i,\tau}$ is the estimated advantage. Gradients are backpropagated not only through the policy $\tilde{\pi}_i$ but also through the sampling distribution $\rho(G_{\mathcal{V}_i} \mid s_{\mathcal{V}_i})$, enabling joint optimization of action selection and interaction structure.*

### 4.2 BayesG: Variational Latent Ego-graph Inference for Policy Optimisation

Recall that the graph distribution $\rho(G_{\mathcal{V}_i} \mid s_{\mathcal{V}_i})$, introduced in Definition 3, governs which neighbors agent $i$ attends to. To infer this stochastic distribution over the fixed physical topology, we model it via a binary mask $Z_i \in \{0,1\}^{|\mathcal{V}_i|\times|\mathcal{V}_i|}$ applied over the physical neighborhood graph $G^{env}_{\mathcal{V}_i}$, where each entry $z_{ij} = 1$ indicates that agent $i$ communicates with neighbor $j$. This yields the effective subgraph:

$$G^*_i = Z_i \odot G^{env}_{\mathcal{V}_i}. \tag{6}$$

We formalise the learning of the latent edge mask $Z_i$ as a Bayesian inference problem. This is a natural choice as Bayes' theorem offers a principled way to infer a latent distribution conditioned on the directly observable physical graph $G_{\mathcal{V}_i}^{\text{env}}$ and agent-local data $D_i$. Treating edge masks as latent variables enables agents to capture uncertainty over coordination structures—particularly valuable when noisy or non-stationary dynamics make deterministic graph pruning unreliable.

Let $D_i$ denote agent-local data collected during training, such as neighbor states, policy outputs, and trajectory embeddings. The posterior distribution over edge masks is given by Bayes' theorem:

$$p(Z_i \mid G_{\mathcal{V}_i}^{\text{env}}, D_i) \propto p(D_i \mid Z_i, G_{\mathcal{V}_i}^{\text{env}}) \cdot p(Z_i), \tag{7}$$

where $p(D_i \mid Z_i, G_{\mathcal{V}_i}^{\text{env}})$ is the likelihood, measuring how well the masked graph explains observed behavior, and $p(Z_i)$ is a prior (e.g., Bernoulli with bias).

To enable tractable optimization, we approximate the posterior with a variational distribution:

$$q(Z_i; \phi_i) = \prod_{j \in \mathcal{N}_i} \text{Bern}(z_{ij}; \sigma(\phi_{ij})), \tag{8}$$

where $\phi_i$ are learnable logits and $\sigma(\cdot)$ is the sigmoid function. We apply the Gumbel-Softmax reparameterization[35, 36] to allow gradient-based learning through discrete edge sampling.

The variational parameters $\phi_i$ are learned by minimizing the Kullback–Leibler (KL) divergence between the approximate posterior $q(Z_i; \phi_i)$ and the true Bayesian posterior $p(Z_i \mid G_{\mathcal{V}_i}^{\text{env}}, D_i)$:

$$\text{KL}[q(Z_i; \phi_i) \| p(Z_i \mid G_{\mathcal{V}_i}^{\text{env}}, D_i)] = \mathbb{E}_{q(Z_i)}\left[\log q(Z_i) - \log p(Z_i \mid G_{\mathcal{V}_i}^{\text{env}}, D_i)\right]. \tag{9}$$

This approximation enables each agent to learn a structured distribution over interaction graphs that reflects the most informative subgraphs for policy optimization. Following the standard variational inference framework, minimizing this KL divergence is equivalent (up to an additive constant) to maximizing the evidence lower bound (ELBO), or equivalently, minimizing the negative ELBO:

$$\begin{aligned}
\mathcal{L}_{\text{ELBO}} &= \mathbb{E}_{q(Z_i;\phi_i)}\left[\log p(D_i \mid Z_i, G_{\mathcal{V}_i}^{\text{env}}) + \log p(Z_i) - \log q(Z_i; \phi_i)\right] + \text{const} \\
&= \mathbb{E}_{q(Z_i;\phi_i)}\left[-\mathcal{L}_{\theta,\varphi} + \log p(Z_i) - \log q(Z_i; \phi_i)\right] + \text{const},
\end{aligned} \tag{10}$$

where the likelihood term $\log p(D_i \mid Z_i, G_{\mathcal{V}_i}^{\text{env}})$ is instantiated by the graph-conditioned actor loss $\mathcal{L}_{\theta,\varphi}$ defined in Definition 4, which provides task feedback to guide the learning of latent structures. (See Appendix A.3 for details.)

**Connection to maximum entropy RL.** Our formulation parallels the probabilistic interpretation of RL in Soft Actor-Critic (SAC) [37], which casts policy optimization as entropy-regularized inference. In SAC, policy entropy $\mathcal{H}(\pi(\cdot|s))$ regularizes action selection; in our framework, the term $-\log q(Z_i; \phi_i)$ in Equation (10) acts as mask entropy $\mathcal{H}(q(Z_i; \phi_i))$, regularizing the variational distribution over communication structures. This prevents premature collapse to deterministic graphs and enables uncertainty-aware neighbor selection. The dual entropy regularization—over both actions (embedded in $\mathcal{L}_{\theta,\varphi}$) and edges (the $-\log q$ term)—is key to our method's robustness in dynamic environments where action diversity and adaptive communication are both critical.

Expanding the ELBO in Equation (10), we obtain:

**Definition 5 (BayesG-ELBO Objective).** *The BayesG objective integrates policy learning and graph inference in a unified variational framework. For agent $i$, the ELBO is:*

$$\mathcal{L}_{\text{ELBO}} = \mathbb{E}_{q(Z_i;\phi_i)}\left[-\mathcal{L}_{\theta,\varphi} + \sum_{j \in \mathcal{N}_i}\left((\lambda + \sigma(\phi_{ij}))\log\sigma(\phi_{ij}) + (2 - \lambda - \sigma(\phi_{ij}))\log(1 - \sigma(\phi_{ij}))\right)\right], \tag{11}$$

*where $\mathcal{L}_{\theta,\varphi}$ denotes the graph-conditioned policy loss defined in Definition 4, and $\sigma(\phi_{ij})$ is the Bernoulli parameter for each candidate edge. This objective jointly optimizes the policy and the structure of the latent subgraph via variational inference.*

We complement the actor-based ELBO with a standard A2C critic loss:

$$\mathcal{L}_{\text{Critic}} = \frac{1}{|\mathcal{B}|}\sum_{\tau \in \mathcal{B}}\left(V_{\omega_i}(\tilde{s}_{i,\tau}, u_{\mathcal{N}_i,\tau}) - \hat{R}_{i,\tau}^{\pi}\right)^2, \tag{12}$$

where $\tilde{s}_{i,\tau}$ is the graph-conditioned input for the critic, and $\hat{R}_{i,\tau}^{\pi}$ is the estimated return. The final training objective combines both components across agents: $\mathcal{L}_{\text{total}} = \sum_{i \in \mathcal{V}} \left( -\mathcal{L}_{\text{ELBO}} + \mathcal{L}_{\text{Critic}} \right)$. This formulation enables decentralized agents to learn both communication structure and policy behavior from local interactions, while adhering to the physical structure of the environment. The ego-graph terminology reflects that each agent independently infers its own local communication neighborhood. Full derivations are provided in Appendix A.3.

## 5 Experiments

### 5.1 Experimental Setup

We evaluate BayesG on five benchmark scenarios for adaptive traffic signal control (ATSC), implemented using the SUMO microscopic traffic simulator [38]. Each environment simulates peak-hour traffic, with one MDP step corresponding to a fixed control interval. Agents observe traffic conditions via induction-loop detectors (ILDs), including vehicle density, queue lengths, and waiting times on incoming lanes, and control local traffic signals. The reward is the negative number of halted vehicles, normalized by a fixed scale. These environments naturally satisfy Definition 1: state transitions depend on local and neighbor actions, rewards are localized per agent, and neighborhood structure is fixed by the road network. Detailed MDP mappings are in Appendix B. [1]

**ATSC_Grid and Monaco.** These two medium-scale scenarios are widely used in prior ATSC benchmarks (e.g., NeurComm [7]). `ATSC_Grid` is a synthetic $5 \times 5$ network with regular connectivity, while `Monaco` replicates a real-world 28-intersection layout. Both use a 5-second control interval and run for 720 steps per episode, totaling 3600 seconds of simulated time. States include lane-level traffic conditions and neighbor observations. Actions correspond to local phase switches, and a 2-second yellow phase is enforced for safety.

**NewYork.** To assess scalability, we introduce three large-scale scenarios—`NewYork33`, `NewYork51`, and `NewYork167`—comprising 33, 51, and 167 signalized intersections, respectively. These networks are derived from real-world Manhattan layouts (see Appendix C for more details). We use a 20-second control interval (increased to 40 seconds for `NewYork167`) and simulate 500 steps per episode. Intersections feature heterogeneous phase designs and ILDs configurations. States include normalized lane-level metrics and neighborhood-aware features. These environments pose significant challenges for coordination and generalization in decentralized MARL.

#### 5.1.1 Baselines

We compare **BayesG** against six representative multi-agent actor–critic baselines, all implemented using a unified A2C backbone with consistent neighbor access and decentralized execution:

**IA2C** [13]: Independent actor–critic training where each agent optimizes its policy based on local observations and a critic that observes neighboring actions. No inter-agent communication is used.

**ConseNet** [16]: A consensus-based method where critics synchronize neighbours' parameters via local averaging to promote stability and coordination, without exchanging message content.

**FPrint** [39]: Mitigates non-stationarity by attaching policy fingerprints to local transitions, enabling agents to condition their updates on the behavior of neighbors over time.

**LToS** [19]: Introduces a hierarchical reward-sharing scheme where agents learn to shape local rewards via neighborhood-level value estimates. No explicit communication is involved.

**CommNet** [40]: A communication-based architecture where each agent receives the average of all encoded neighbor messages, limiting expressiveness but allowing simple message integration.

**NeurComm** [7]: A more flexible communication protocol that encodes and aggregates messages from neighbors without averaging, supporting richer interaction modeling. It generalizes earlier approaches such as CommNet and DIAL [41].

We categorize **IA2C**, **ConseNet**, **FPrint**, and **LToS** as *non-communicative* baselines, as they do not involve direct message exchange. In contrast, **CommNet**, **NeurComm**, and our method **BayesG** are *communication-based*.

---

[1] Code and data are available at `https://github.com/Wei9711/BayesG`.

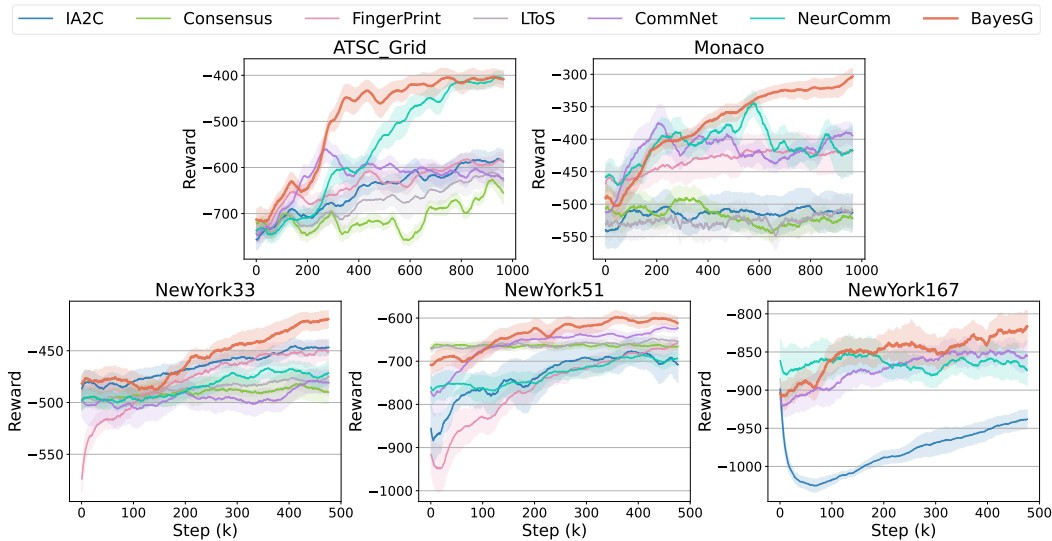

Figure 2: Training reward curves of BayesG and baselines across five ATSC environments. BayesG consistently achieves higher returns and faster convergence, particularly in large-scale settings, demonstrating the benefit of learning task-adaptive communication graphs.

### 5.1.2 Implementation Details

For all environments, we use a fixed control interval of 5 seconds. Policy and critic networks share similar architectures across all baselines. Each experiment is averaged over 5 random seeds.

To highlight the differences of communication-based methods, we describe how each method constructs the agent-specific input representation $\tilde{s}_i$, which is input to the policy and critic networks. All methods encode combinations of the following components: (1) **State features** $(s_i, s_{\mathcal{N}_i})$: capturing local and neighboring state; (2) **Policy features** $(\pi_{\mathcal{N}_i})$: representing the action distributions (fingerprints) of neighboring agents. (3) **Trajectory features** $(h_{\mathcal{N}_i})$: represented as the hidden states of RNNs that reflect recent behavior. Each channel is encoded via a multilayer perceptron (MLP), unless otherwise noted.

**CommNet.** Each agent encodes its own and neighboring observations with an MLP, and aggregates neighbors' trajectory features via mean pooling followed by another MLP. The final input is the sum of these two components: $\tilde{s}_i = \mathrm{MLP}_{\mathrm{state}}([s_i, s_{\mathcal{N}_i}]) + \mathrm{MLP}_{\mathrm{traj}}(\mathrm{mean}(h_{\mathcal{N}_i}))$.

**NeurComm.** NeurComm encodes each of the three information types separately and concatenates their embeddings before passing the result into an LSTM for temporal reasoning: $\tilde{s}_i = \mathrm{MLP}_{\mathrm{state}}([s_i, s_{\mathcal{N}_i}]) \parallel \mathrm{MLP}_{\mathrm{policy}}(\pi_{\mathcal{N}_i}) \parallel \mathrm{MLP}_{\mathrm{traj}}(h_{\mathcal{N}_i})$.

**BayesG (Ours).** BayesG replaces fixed communication with a learned latent subgraph. Let $A_i \in \{0,1\}^{|\mathcal{V}_i| \times |\mathcal{V}_i|}$ denote the local physical graph and $Z_i \in \{0,1\}^{|\mathcal{V}_i| \times |\mathcal{V}_i|}$ a sampled binary mask; the effective graph is defined as: $A_i^* = Z_i \odot A_i$. The input is then computed via masked message passing over $A_i^*$ using GNNs for each input channel:

$$\tilde{s}_i = \mathrm{GNN}_{\mathrm{obs}}(S_{\mathcal{V}_i}, A_i^*) \parallel \mathrm{GNN}_{\mathrm{policy}}(\Pi_{\mathcal{V}_i}, A_i^*) \parallel \mathrm{GNN}_{\mathrm{traj}}(H_{\mathcal{V}_i}, A_i^*). \tag{13}$$

Here, $S_{\mathcal{V}_i}$, $\Pi_{\mathcal{V}_i}$, and $H_{\mathcal{V}_i}$ represent the observation, policy, and trajectory features of agent $i$'s neighborhood. While any GNN can be employed, we use graph convolutional networks (GCNs) [42] for efficiency in our implementation. This formulation allows each agent to learn sparse and context-aware communication structures tailored to local dynamics. (See pseudocode in Appendix D).

### 5.2 Performance Comparison

**Training Performance.** Figure 2 presents the training curves of BayesG and baseline methods across five ATSC environments. Each curve shows the smoothed average episode return (defined as the

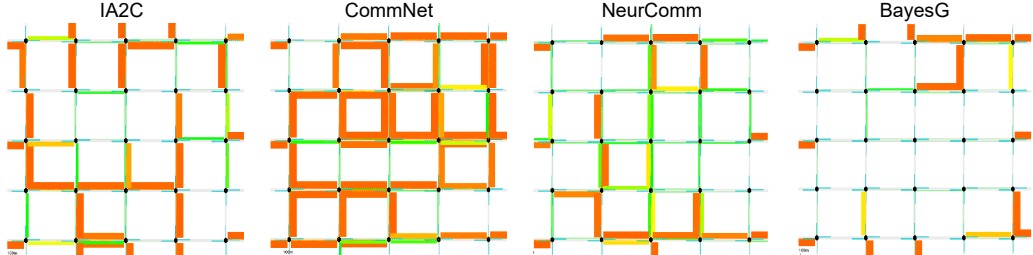

Figure 3: Visualization of traffic congestion on the Grid map at 3500 simulation seconds. Road thickness and color represent vehicle density (thicker and redder indicates more congestion). BayesG results in significantly lighter and sparser traffic flows.Additional qualitative comparisons across different simulation times are provided in Figure 10 (Appendix F).

negative of total queue length) over 5 random seeds. Several key trends emerge: **(1) Consistent superiority of BayesG**: Across all environments, BayesG steadily outperforms both non-communicative baselines (IA2C, ConseNet, FPrint, LToS) and explicit communication protocols (CommNet, NeurComm). This highlights the advantage of learning task-adaptive latent subgraphs that promote efficient and stable coordination. **(2) Faster convergence**: BayesG exhibits faster convergence in early training, particularly on `ATSC_Grid` and `Monaco`. This indicates that selective interaction via learned masks accelerates policy learning by reducing noisy or redundant communication. **(3) Scalability in large networks**: On the large-scale `NewYork33`, `NewYork51`, and `NewYork167` scenarios, BayesG achieves higher asymptotic returns and more stable learning curves, while baselines often suffer from plateauing or instability. This validates BayesG's ability to scale to high-dimensional decentralized settings by suppressing uninformative edges. These results collectively demonstrate that BayesG not only enhances learning efficiency but also improves final policy quality by discovering sparse, context-aware coordination patterns.

To further understand the optimization dynamics of BayesG, we report the evolution of key training loss components in Appendix E for `ATSC_Grid`, `Monaco`, and `NewYork33`. Specifically, we track the policy loss, value (critic) loss, ELBO loss, prior loss, mask entropy regularization, and the total objective. These loss curves offer deeper insight into how BayesG balances policy optimization with variational graph inference, and demonstrate the stability of the joint training process across different network scales.

**Qualitative Comparison on Grid Environment**. Figure 3 presents a visual comparison of traffic congestion for IA2C, CommNet, NeurComm, and BayesG on the `ATSC_Grid` map at 3,500 simulation seconds. Road segment thickness and color indicate vehicle density, with thicker and redder lines denoting heavier congestion. IA2C and CommNet exhibit severe congestion across many intersections, while NeurComm reduces some bottlenecks through explicit communication. BayesG, in contrast, achieves the smoothest traffic flow, with thinner and less congested segments. These results highlight BayesG's ability to adaptively focus on critical interactions via its learned latent graph, enabling more effective and scalable coordination in dense traffic scenarios. Additional qualitative comparisons across different simulation times are provided in Appendix F.

### 5.3 Case Study: Interpreting Learned Interaction Structures

To further understand the behavior of BayesG, we conduct a case study on the `ATSC_Grid` map at time step 1,400. Figure 4 illustrates: **(a) Left:** The latent interaction matrix, where each row corresponds to an agent's inferred ego-graph—i.e., a probabilistic mask over its physical neighborhood. The aggregated matrix represents the likelihood of interaction between all agent pairs.**(a) Right:** The physical road network layout with edges colored and weighted according to the same communication probabilities, providing an interpretable spatial view. **(b):** The vehicle density snapshot from SUMO simulation at the same time step. Thicker and redder road segments indicate higher congestion levels.

This visualization shows that BayesG learns to prioritize communication from low-congestion intersections toward more congested regions. For example, when congestion builds up in a particular area (e.g., top-right and bottom right clusters), left-stream intersections increase their coordination

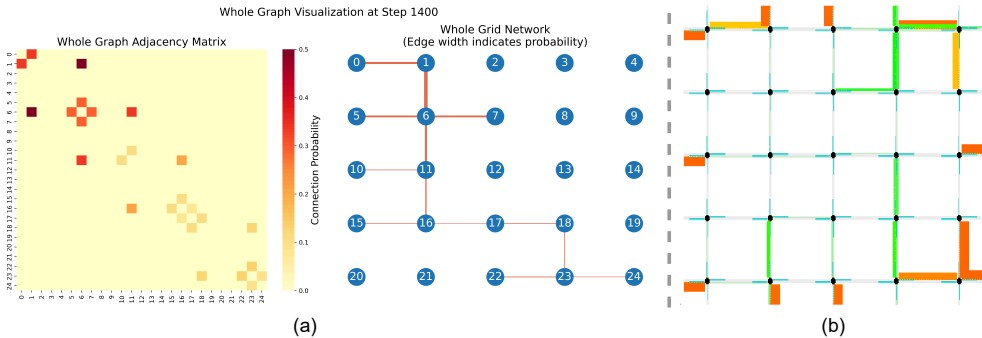

Figure 4: Case study on the `ATSC_Grid` map at time step 1400. **(a)** The latent interaction graph inferred by BayesG. Each agent samples a probabilistic binary mask over its ego-graph and the global latent graph is formed by aggregating these per-agent ego-graph masks. Edge thickness reflects the inferred likelihood of communication between intersections. **(b)** The vehicle density snapshot from the SUMO simulation. Thicker red/orange lines denote higher vehicle accumulation on road segments, indicating local congestion.

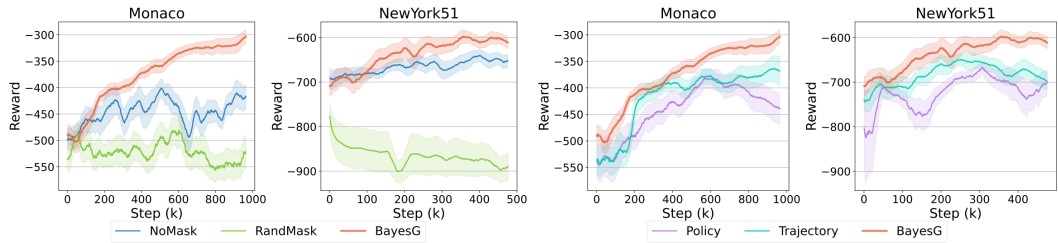

Figure 5: **Ablation study on the `Monaco` and `NewYork51` environments. Left:** Performance comparison of different graph masking strategies. **Right:** Effect of different feature types used to generate the variational mask.

weights, effectively regulating inflow and providing more green phases to help alleviate downstream pressure. Unlike approaches that treat all neighbors equally, BayesG captures directional, task-driven cooperation: it encourages agents outside a congested zone to adjust proactively, thereby relieving bottlenecks before they worsen. This reflects real-world traffic control principles, where strategic upstream coordination plays a critical role in minimizing congestion spread. These results demonstrate BayesG's ability to infer decentralized, context-aware communication structures that adapt dynamically to evolving traffic conditions. (See more cases in Appendix G)

### 5.4 Ablation Studies

To better understand the impact of BayesG's components, we conduct two sets of ablation studies on the `Monaco` and `NewYork51` environments, focusing on (i) how the graph mask is generated, and (ii) what input features are used to learn the mask. Results are summarized in Figure 5.

**Graph Masking Strategies.** We compare three strategies for subgraph construction: (1) **No Masking**: Uses the raw environment topology $A_i$ without any pruning. (2) **Random Masking**: Applies a randomly sampled binary mask $Z_i$, simulating naive edge dropout. (3) **Learned Mask (BayesG)**: Applies the full variational inference pipeline to infer $Z_i$, resulting in adaptive and task-specific subgraphs. As shown in Figure 5 (left), random masking severely degrades performance, while the learned mask significantly outperforms both baselines by suppressing irrelevant links and preserving only informative neighbor connections.

**Mask Input Features.** We further examine how different information types affect the learned mask. Specifically, we use:(1) **State**: Mask is generated based on neighbor observations $s_{\mathcal{V}_i}$. (2) **Trajectory**: Uses LSTM hidden states $h_{\mathcal{V}_i}$, capturing temporal behavior. (3) **Policy**: Based on neighbor action distributions $\pi_{\mathcal{N}_i}$. Results in Figure 5 (right) indicate that trajectory-based features lead to stronger performance that state-only, as the incorporate richer temporal context. Policy-based masks also provide meaningful signals, reflecting agents' behavioral intentions. The best results are achieved when the learned mask uses a combination of these features, as implemented in BayesG.

## 6 Discussion

### 6.1 Comparison with Sampling-based Networked MARL

Recent work has explored sampling strategies for scalability in networked MARL, but differs fundamentally in what is sampled, how distributions are determined, and where adaptivity occurs.

**Sampling targets and distributions.** Qu et al. [26] establish convergence results for networked systems with local dependencies on fixed graphs. Lin et al. [23] sample active links from a predefined distribution $\mathcal{D}$ for $\mu$-decay analysis. Anand and Qu [27] use uniform agent subsampling ($k$ out of $n$) for a global controller. In contrast, our method learns context-dependent, per-edge distributions $q(Z_i; \phi_i)$ optimized jointly with the policy via an ELBO.

**Locus of adaptivity.** Lin et al. [23] achieve dynamics-level stochasticity through time-varying link sets. Anand and Qu [27] vary agent subsets for a global controller. Our approach operates at the policy level: each agent samples a task-adaptive 1-hop subgraph for local decision-making.

**First-hop inclusion probabilities.** Lin et al. [23] use the marginal of $\mathcal{D}$ (predefined, not state-adaptive). Anand and Qu [27] use uniform inclusion ($k/n$). Our method learns edge-specific probabilities $\Pr[z_{ij} = 1] = \sigma(\phi_{ij}(\cdot))$ that are state and trajectory-dependent.

These approaches provide valuable convergence guarantees under structural assumptions. Our contribution is complementary: learning which neighbors to attend to, optimized end-to-end with the policy. A detailed comparison is in Table 2 (Appendix H).

### 6.2 Distinction from Dec-POMDP-based Coordination Graphs

Our method addresses the Spatiotemporal-MDP setting (Definition 1), where agent transitions $p_i(s'_i|s_{\mathcal{V}_i}, u_i, u_{\mathcal{N}_i})$ and rewards $R(s_{\mathcal{V}_i}, u_{\mathcal{V}_i})$ are localized over physically connected neighbors determined by $G_{\text{env}}$. This reflects real-world systems (traffic networks, power grids, sensor fields) where agents only observe and affect local neighborhoods through fully decentralized learning.

In contrast, Dec-POMDP-based methods such as CASEC [15], DCG [33], DICG [29], SOP-CG [34], and GACG [22] assume global state $s$, global reward $R(s, \mathbf{u})$, joint transitions $p(s'|s, \mathbf{u})$, and centralized training with decentralized execution (CTDE). They allow unrestricted graph rewiring unconstrained by physical topology, making them incompatible with physically grounded networked systems. Our baselines—CommNet [40], NeurComm [7], and LToS [23]—communicate only with physical neighbors, respect fixed topology, and perform fully decentralized learning. While Dec-POMDP coordination graphs are important contributions, we exclude them due to incompatibility with Spatiotemporal-MDP's decentralized, physically constrained nature.

## 7 Conclusion

We presented **BayesG**, a variational framework for learning latent interaction graphs in networked multi-agent reinforcement learning. By operating over *ego-graphs*—localized subgraphs centered on each agent—BayesG enables decentralized agents to infer task-adaptive communication structures using local observations and Bayesian inference. Our approach integrates graph inference with actor–critic training via an ELBO objective, jointly optimizing both the interaction topology and policy. Experiments on five adaptive traffic signal control benchmarks demonstrate that BayesG consistently outperforms strong baselines, achieving superior coordination and reduced congestion. Case studies and ablation results further highlight the efficiency and interpretability of the learned graphs. BayesG offers a principled approach to structure-aware cooperation in decentralized settings and opens up new opportunities for scalable learning in networked systems.

## Acknowledgements

This work is supported by the Australian Research Council under Australian Laureate Fellowships FL190100149.

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

# Appendix

## A    Detailed Derivation

### A.1    Derivation of Graph-based Policy

We expand on Definition 3, which defines each agent's policy as a two-stage process: (i) sampling a binary subgraph $G_{\mathcal{V}_i}$ from a variational distribution $\rho$, and (ii) conditioning the action policy on the graph-filtered encoding $\tilde{f}_i(s_{\mathcal{V}_i}, G_{\mathcal{V}_i})$.

In conventional MARL, each agent conditions its policy on a fixed observation $\tilde{s}_i = f(s_{\mathcal{V}_i})$, where $\mathcal{V}_i$ is the agent's closed neighborhood. In our formulation, the observation additionally depends on a latent subgraph $G_{\mathcal{V}_i}$ drawn from a learned distribution:

$$G_{\mathcal{V}_i} \sim \rho(G \mid s_{\mathcal{V}_i}; \varphi_i), \tag{14}$$

where $\rho$ is parameterized by variational parameters $\varphi_i$. To ensure feasibility, we constrain sampled subgraphs to lie within the physical topology:

$$G_{\mathcal{V}_i} \preceq G_{\mathcal{V}_i}^{\text{env}}, \tag{15}$$

i.e., agents can only sample from edges permitted by the environment graph.

The graph-conditioned encoder $\tilde{f}_i(s_{\mathcal{V}_i}, G_{\mathcal{V}_i})$ is implemented using GNNs over the masked subgraph. The resulting policy is defined as:

$$\pi_i(u_i \mid s_{\mathcal{V}_i}) = \mathbb{E}_{G_{\mathcal{V}_i} \sim \rho} \left[ \tilde{\pi}_i(u_i \mid \tilde{f}_i(s_{\mathcal{V}_i}, G_{\mathcal{V}_i}); \theta_i) \right], \tag{16}$$

where $\tilde{\pi}_i$ is the action policy given graph-filtered input.

This formulation enables each agent to dynamically adapt its local observation space via sampled interaction subgraphs, supporting sparse and context-aware reasoning under topological constraints.

### A.2    Derivation of Graph-based A2C Objective

We derive the actor loss introduced in Definition 4, which integrates latent subgraph sampling into A2C learning.

Recall the standard A2C actor loss:

$$\mathcal{L}^{\text{A2C}}(\theta_i) = -\log \pi_{\theta_i}(u_i \mid \tilde{s}_i) \cdot \hat{A}_i^\pi, \tag{17}$$

where $\tilde{s}_i$ is the observation, $u_i$ the action, and $\hat{A}_i^\pi$ the estimated advantage.

Under the graph-based policy framework, the policy depends on a sampled subgraph $G_{\mathcal{V}_i}$:

$$\pi_i(u_i \mid s_{\mathcal{V}_i}) = \mathbb{E}_{G_{\mathcal{V}_i} \sim \rho(\cdot \mid s_{\mathcal{V}_i}; \varphi_i)} \left[ \tilde{\pi}_i(u_i \mid \tilde{f}_i(s_{\mathcal{V}_i}, G_{\mathcal{V}_i}); \theta_i) \right]. \tag{18}$$

The expected actor loss becomes:

$$\mathcal{L}_{\theta,\varphi} = \mathbb{E}_{G_{\mathcal{V}_i} \sim \rho} \left[ -\log \tilde{\pi}_i(u_i \mid \tilde{f}_i(s_{\mathcal{V}_i}, G_{\mathcal{V}_i})) \cdot \hat{A}_i^\pi \right]. \tag{19}$$

We add entropy regularization to encourage exploration:

$$\mathcal{H}(\tilde{\pi}_i(\cdot \mid \tilde{f}_i)) = -\sum_{u_i \in \mathcal{U}^i} \tilde{\pi}_i(u_i \mid \tilde{f}_i) \log \tilde{\pi}_i(u_i \mid \tilde{f}_i). \tag{20}$$

Over a batch $\mathcal{B}$, the full objective is:

$$\mathcal{L}_{\theta,\varphi} = \frac{1}{|\mathcal{B}|} \sum_{\tau \in \mathcal{B}} \mathbb{E}_{G_{\mathcal{V}_i} \sim \rho} \left[ -\log \tilde{\pi}_i(a_{i,\tau} \mid \tilde{f}_i(s_{\mathcal{V}_i}, G_{\mathcal{V}_i})) \cdot \hat{A}_{i,\tau}^\pi + \beta \cdot \mathcal{H}(\tilde{\pi}_i(\cdot \mid \tilde{f}_i)) \right]. \tag{21}$$

This defines the graph-based actor loss used in our main training objective.

## A.3 Derivation of Variational Objective and ELBO for BayesG

We present the detailed derivation of the ELBO objective used in BayesG, beginning with Bayes' theorem and ending with a tractable training objective for policy and graph optimization.

### Bayesian Inference over Latent Graphs

In networked MARL, the environment graph $G_{\mathcal{V}_i}^{\text{env}}$ represents the physical connectivity between agents (e.g., traffic lights connected via roads), which imposes hard topological constraints on interaction. We model the stochastic interaction graph for agent $i$ as a binary mask $Z_i \in \{0, 1\}^{|\mathcal{V}_i| \times |\mathcal{V}_i|}$, sampled from a learned distribution over the physical neighborhood graph $G_{\mathcal{V}_i}^{\text{env}}$. The resulting posterior distribution is:

$$p(Z_i \mid G_{\mathcal{V}_i}^{\text{env}}, D_i) = \frac{p(D_i \mid Z_i, G_{\mathcal{V}_i}^{\text{env}}) \cdot p(Z_i)}{p(D_i \mid G_{\mathcal{V}_i}^{\text{env}})}, \tag{22}$$

where:

- $D_i$ denotes the agent-specific data (e.g., neighbor states, trajectories, and policies),
- $p(Z_i)$ is a prior over edge masks,
- $p(D_i \mid Z_i, G_{\mathcal{V}_i}^{\text{env}})$ is the likelihood of the data under the masked graph,
- $p(D_i \mid G_{\mathcal{V}_i}^{\text{env}})$ is the marginal likelihood.

### Variational Approximation and KL Divergence

Since the exact posterior in Eq. (22) is intractable, we introduce a variational distribution $q(Z_i; \phi_i)$ to approximate it. The variational parameters $\phi_i$ are learned by minimizing the Kullback–Leibler (KL) divergence between the approximate posterior and the true posterior:

$$\text{KL}[q(Z_i; \phi_i) \| p(Z_i \mid G_{\mathcal{V}_i}^{\text{env}}, D_i)] = \mathbb{E}_{q(Z_i)} \left[ \log q(Z_i) - \log p(Z_i \mid G_{\mathcal{V}_i}^{\text{env}}, D_i) \right]. \tag{23}$$

Applying Bayes' rule to the log posterior, we rewrite:

$$\log p(Z_i \mid G_{\mathcal{V}_i}^{\text{env}}, D_i) = \log p(D_i \mid Z_i, G_{\mathcal{V}_i}^{\text{env}}) + \log p(Z_i) - \log p(D_i \mid G_{\mathcal{V}_i}^{\text{env}}). \tag{24}$$

Substituting Eq. (24) into Eq. (23) yields:

$$\text{KL} = \mathbb{E}_{q(Z_i)} \left[ \log q(Z_i) - \log p(D_i \mid Z_i, G_{\mathcal{V}_i}^{\text{env}}) - \log p(Z_i) + \log p(D_i \mid G_{\mathcal{V}_i}^{\text{env}}) \right] \tag{25}$$

$$= \mathbb{E}_{q(Z_i)} \left[ \log q(Z_i) - \log p(D_i \mid Z_i, G_{\mathcal{V}_i}^{\text{env}}) - \log p(Z_i) \right] + \log p(D_i \mid G_{\mathcal{V}_i}^{\text{env}}) \tag{26}$$

$$= -\mathbb{E}_{q(Z_i)} \left[ \log p(D_i \mid Z_i, G_{\mathcal{V}_i}^{\text{env}}) + \log p(Z_i) - \log q(Z_i) \right] + \log p(D_i \mid G_{\mathcal{V}_i}^{\text{env}}) \tag{27}$$

$$= -\mathbb{E}_{q(Z_i)} \left[ \log p(D_i \mid Z_i, G_{\mathcal{V}_i}^{\text{env}}) + \log p(Z_i) - \log q(Z_i) \right] + \text{const}, \tag{28}$$

where we have used the fact that $\log p(D_i \mid G_{\mathcal{V}_i}^{\text{env}})$ does not depend on $Z_i$ or $\phi_i$. Therefore, minimizing the KL divergence is equivalent (up to an additive constant) to maximizing the evidence lower bound (ELBO):

$$\mathcal{L}_{\text{ELBO}} = \mathbb{E}_{q(Z_i; \phi_i)} \left[ \log p(D_i \mid Z_i) + \log p(Z_i) - \log q(Z_i; \phi_i) \right] + \text{const}. \tag{29}$$

### Term-by-Term Breakdown

**1. Likelihood Term**: $\log p(D_i \mid Z_i)$

This is modeled using the graph-conditioned policy loss, i.e.,

$$\log p(D_i \mid Z_i) \approx -\mathcal{L}_{\theta, \varphi}, \tag{30}$$

where $\mathcal{L}_{\theta, \varphi}$ is the actor loss under sampled subgraph $Z_i \odot G_{\mathcal{V}_i}^{\text{env}}$.

**2. Prior Term**: $\log p(Z_i)$

We define the prior as an element-wise Bernoulli with retention bias $\lambda$:

$$p(Z_i) = \prod_{j \in \mathcal{N}_i} \lambda^{z_{ij}} (1 - \lambda)^{1 - z_{ij}}. \tag{31}$$

Then:

$$\log p(Z_i) = \sum_{j \in \mathcal{N}_i} z_{ij} \log \lambda + (1 - z_{ij}) \log(1 - \lambda). \tag{32}$$

Taking expectation under $q(Z_i)$:

$$\mathbb{E}_q[\log p(Z_i)] = \sum_{j \in \mathcal{N}_i} \left[ \sigma(\phi_{ij}) \log \lambda + (1 - \sigma(\phi_{ij})) \log(1 - \lambda) \right]. \tag{33}$$

**3. Entropy Term**: $-\log q(Z_i; \phi_i)$

Since $q(Z_i)$ is a factorized Bernoulli:

$$H(q(Z_{ij})) = -\sigma(\phi_{ij}) \log \sigma(\phi_{ij}) - (1 - \sigma(\phi_{ij})) \log(1 - \sigma(\phi_{ij})). \tag{34}$$

Then:

$$\mathbb{E}_q[\log q(Z_i)] = -\sum_{j \in \mathcal{N}_i} H(q(Z_{ij})). \tag{35}$$

**Final Objective**

Combining all terms:

$$
\begin{aligned}
\mathcal{L}_{\text{ELBO}} &= \mathbb{E}_{q(Z_i; \phi_i)}\left[ -\mathcal{L}_{\theta,\varphi} \right] + \sum_{j \in \mathcal{N}_i} \left[ \lambda \log \sigma(\phi_{ij}) + (1 - \lambda) \log(1 - \sigma(\phi_{ij})) \right] - \sum_{j \in \mathcal{N}_i} H(q(Z_{ij})) \\
&= \mathbb{E}_{q(Z_i; \phi_i)}\left[ -\mathcal{L}_{\theta,\varphi} \right] + \sum_{j \in \mathcal{N}_i} \left[ \lambda \log \sigma(\phi_{ij}) + (1 - \lambda) \log(1 - \sigma(\phi_{ij})) \right] \\
&\qquad\qquad\qquad - \sum_{j \in \mathcal{N}_i} \left[ -\sigma(\phi_{ij}) \log \sigma(\phi_{ij}) - (1 - \sigma(\phi_{ij})) \log(1 - \sigma(\phi_{ij})) \right] \\
&= \mathbb{E}_{q(Z_i; \phi_i)}\Bigg[ -\mathcal{L}_{\theta,\varphi} + \sum_{j \in \mathcal{N}_i} \Big( \lambda \log \sigma(\phi_{ij}) + (1 - \lambda) \log(1 - \sigma(\phi_{ij})) \\
&\qquad\qquad\qquad\qquad + \sigma(\phi_{ij}) \log \sigma(\phi_{ij}) + (1 - \sigma(\phi_{ij})) \log(1 - \sigma(\phi_{ij})) \Big) \Bigg] \\
&= \mathbb{E}_{q(Z_i; \phi_i)}\Bigg[ -\mathcal{L}_{\theta,\varphi} + \sum_{j \in \mathcal{N}_i} \big( (\lambda + \sigma(\phi_{ij})) \log \sigma(\phi_{ij}) + (2 - \lambda - \sigma(\phi_{ij})) \log(1 - \sigma(\phi_{ij})) \big) \Bigg],
\end{aligned}
\tag{36}
$$

This is the BayesG training objective as formalized in Definition 5.

## B  ATSC Environment: MDP Component Mapping and Spatiotemporal-MDP Justification

We provide detailed mappings from the adaptive traffic signal control (ATSC) domain to the Spatiotemporal-MDP framework (Definition 1).

### B.1  MDP Components

**States ($s_i$).** Each agent $i$ (a signalized intersection) observes its local traffic state via induction loop detectors (ILDs) placed on incoming lanes. The state includes:

- **Vehicle density**: Number of vehicles per lane
- **Queue length**: Number of stopped vehicles per lane
- **Waiting time**: Average waiting time of vehicles per lane
- **Current phase**: Active traffic signal phase (e.g., north-south green)

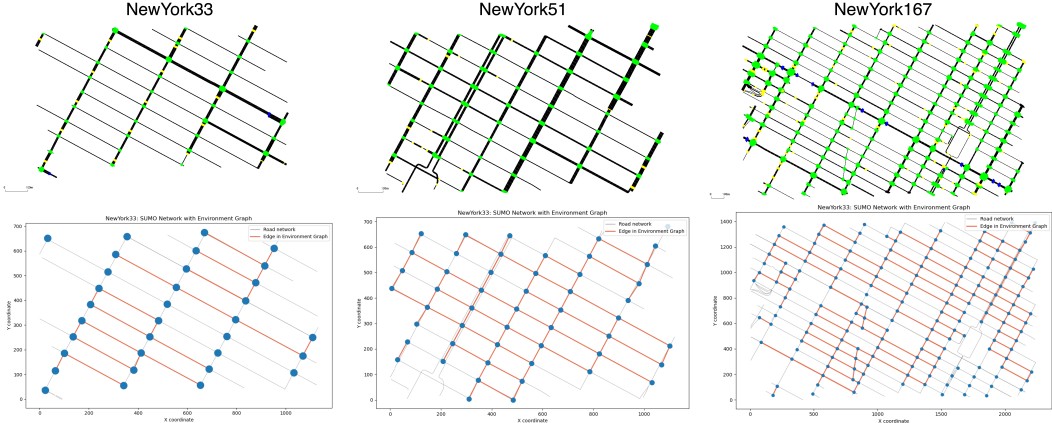

Figure 6: Visualization of the `NewYork33`, `NewYork51`, and `NewYork167` environments. Top: SUMO network with signalized intersections. Bottom: extracted graph structure used in networked MARL, including traffic light nodes and their physical neighbors.

- **Phase duration**: Time elapsed in current phase

For neighborhood-aware coordination, agents also receive aggregated statistics from immediate neighbors $\mathcal{N}_i$ (e.g., neighbor queue lengths, active phases).

**Actions** $(u_i)$**.** Each agent selects a traffic signal phase from a discrete action space. Actions correspond to:

- **Phase switching**: Transition to a different signal phase (e.g., from north-south green to east-west green)
- **Phase holding**: Maintain the current phase for another control interval

A mandatory yellow phase (2 seconds) is enforced between conflicting green phases for safety.

**Transition Probabilities** $(p_i(s_i'|s_{\mathcal{V}_i}, u_i, u_{\mathcal{N}_i}))$**.** Transitions are not analytically available but are governed by the SUMO microscopic traffic simulator [38]. The key property is **locality**: agent $i$'s next state $s_i'$ depends primarily on:

- Its own action $u_i$ (phase decision)
- Local state $s_i$ (current congestion)
- Neighbor actions $u_{\mathcal{N}_i}$ (upstream/downstream traffic release)

This satisfies the Spatiotemporal-MDP assumption that transitions are localized over the physical neighborhood $\mathcal{V}_i$.

**Rewards** $(R(s_{\mathcal{V}_i}, u_{\mathcal{V}_i}))$**.** The reward for agent $i$ is:

$$r_i = -\frac{1}{C} \sum_{\ell \in \text{lanes}_i} \text{num\_halted}(\ell), \tag{37}$$

where $\text{num\_halted}(\ell)$ is the number of stopped vehicles on lane $\ell$, and $C$ is a normalization constant. This encourages local queue minimization. The reward is localized to agent $i$'s intersection, consistent with the Spatiotemporal-MDP framework.

## B.2 Why Spatiotemporal-MDP Fits ATSC

The ATSC domain is a canonical example of Spatiotemporal-MDP due to:

1. **Localized dynamics.** Traffic flow is governed by physical proximity: upstream intersections release vehicles that propagate to downstream intersections. Each agent's state evolution depends on its immediate neighbors' actions, not the global joint action of all agents.

2. **Fixed physical topology.** The road network structure is fixed and sparse, with agents (intersections) only interacting with directly connected neighbors via shared road segments.

3. **Decentralized execution requirement.** In real-world deployments, traffic signals operate independently with limited communication bandwidth. Centralized control is impractical due to:

- **Scalability**: City-scale networks have hundreds of intersections; centralized joint action spaces grow exponentially
- **Communication constraints**: Real-time global state aggregation is infeasible under latency and bandwidth limits
- **Robustness**: Centralized systems are vulnerable to single points of failure

4. **Local observability.** Each intersection has sensors only for its incoming lanes, consistent with the partial observability assumption in Spatiotemporal-MDP.

These properties make ATSC fundamentally different from cooperative benchmarks (e.g., Star-Craft) that assume global rewards, unrestricted communication, and arbitrary coordination graphs. Our method exploits this structure by learning sparse, physically grounded communication masks, enabling scalable and deployable traffic control.

## C  NewYork Scenario Visualization and Graph Statistics

To support the evaluation of BayesG on large-scale environments, we include visualizations and statistics for the three real-world maps: `NewYork33`, `NewYork51`, and `NewYork167`. These maps are derived from SUMO simulations based on real intersections in Manhattan, and are visualized in Figure 6.

Each row in Figure 6 shows:

- **Top:** The SUMO map used for microscopic traffic simulation, where green nodes represent traffic light-controlled intersections.
- **Bottom:** The corresponding environment graph used for MARL training. Blue circles represent agents (traffic lights), and red edges indicate neighbor connections used for policy input.

**Note:** The graph used in our MARL environment is a subgraph of the physical road network. Specifically, we include only intersections that are signal-controlled (i.e., managed by traffic lights), and define edges based on direct traffic flow connections between these nodes. Road segments that connect to unsignalized intersections or that skip over intermediate traffic lights are excluded from the environment graph. This design reflects the decentralized setting in which agents can only communicate and coordinate with neighboring controlled intersections.

| Scenario | # Nodes | # Edges | Avg. Degree | Max Degree |
|----------|---------|---------|-------------|------------|
| NewYork33 | 33 | 56 | 1.70 | 3 |
| NewYork51 | 51 | 125 | 2.45 | 4 |
| NewYork167 | 167 | 384 | 2.30 | 4 |

Table 1: Graph statistics for the NewYork traffic signal control environments. Nodes represent signalized intersections; edges represent physical connectivity between controlled agents.

## D  Algorithms

We provide pseudocode for BayesG training and decentralized execution. Algorithm 1 outlines asynchronous multi-agent training with latent graph inference, following the procedures in Sections 3

and 4. The training loop includes: (i) edge sampling and message propagation, (ii) policy and trajectory updates, (iii) value estimation and environment simulation, and (iv) gradient-based updates of actor, critic, and variational graph parameters.

Algorithm 2 describes decentralized execution: each agent samples a subgraph, encodes its neighborhood, and selects actions in real time without global information.

---

**Algorithm 1** BayesG: Multi-agent A2C Training with Variational Graph Inference

---

1: **Parameter:** $\alpha, \beta, \gamma, T, |\mathcal{B}|, \eta_\theta, \eta_\omega, \eta_\phi$
2: **Result:** $\{\theta_i, \omega_i, \phi_i\}_{i \in \mathcal{V}}$
3: Initialize $s_0, \pi_{-1}, h_{-1}, t \leftarrow 0, k \leftarrow 0, \mathcal{B} \leftarrow \emptyset$
4: **repeat**
5:     **for** $i \in \mathcal{V}$ **do**
6:         Sample $Z_i \sim q(Z_i; \phi_i)$, set $A_i^* \leftarrow Z_i \odot G_{\mathcal{V}_i}^{\text{env}}$
7:         Encode $\tilde{s}_{i,t} \leftarrow \tilde{f}_i(s_{\mathcal{V}_i}, A_i^*)$
8:         Update $h_{i,t} \leftarrow g_{\nu_i}(h_{i,t-1}, \tilde{s}_{i,t})$
9:         Sample $a_{i,t} \sim \pi_{\theta_i}(\cdot \mid h_{i,t})$
10:        Compute $v_{i,t} \leftarrow V_{\omega_i}(h_{i,t}, u_{\mathcal{N}_i,t})$
11:        Execute $a_{i,t}$
12:     **end for**
13:     Simulate $\{s_{i,t+1}, r_{i,t}\}_{i \in \mathcal{V}}$
14:     $\mathcal{B} \leftarrow \mathcal{B} \cup \{(s_{i,t}, \pi_{i,t-1}, a_{i,t}, r_{i,t}, v_{i,t})\}_{i \in \mathcal{V}}$
15:     $t \leftarrow t+1, k \leftarrow k+1$
16:     **if** $t = T$ **then**
17:         Reset $s_0, \pi_{-1}, h_{-1}, t \leftarrow 0$
18:     **end if**
19:     **if** $k = |\mathcal{B}|$ **then**
20:         **for** $i \in \mathcal{V}$ **do**
21:             Estimate $\hat{R}_{i,\tau}^\pi, \hat{A}_{i,\tau}^\pi, \forall \tau \in \mathcal{B}$
22:             Update critic: $\omega_i \leftarrow \omega_i - \eta_\omega \nabla \mathcal{L}_{\text{Critic}}$
23:             Update actor & graph: $\theta_i, \phi_i \leftarrow \theta_i, \phi_i - \eta_\theta \nabla_{\theta_i, \phi_i}(-\mathcal{L}_{\text{ELBO}})$
24:         **end for**
25:         Reset $\mathcal{B} \leftarrow \emptyset, k \leftarrow 0$
26:     **end if**
27: **until** Stop condition is reached

---

**Algorithm 2** BayesG: Multi-agent Execution with Latent Graph Sampling

---

1: **Parameter:** $\{\theta_i, \nu_i, \phi_i\}_{i \in \mathcal{V}}, \Delta t_{\text{comm}}, \Delta t_{\text{control}}$
2: **for** $i \in \mathcal{V}$ **do**
3:     Initialize $h_i \leftarrow 0, \pi_i \leftarrow 0, \{s_j, \pi_j\}_{j \in \mathcal{N}_i} \leftarrow 0$
4:     **repeat**
5:         Observe $s_i$
6:         Sample mask $Z_i \sim q(Z_i; \phi_i)$, compute $A_i^* \leftarrow Z_i \odot G_{\mathcal{V}_i}^{\text{env}}$
7:         Send $s_i, \pi_i$ to neighbors
8:         **for** $j \in \mathcal{N}_i$ **do**
9:             Receive and update $s_j, \pi_j$ within $\Delta t_{\text{comm}}$
10:         **end for**
11:         Construct graph-conditioned input: $\tilde{s}_i \leftarrow \tilde{f}_i(s_{\mathcal{V}_i}, A_i^*)$
12:         Update $h_i \leftarrow g_{\nu_i}(h_i, \tilde{s}_i)$, compute policy $\pi_i \leftarrow \pi_{\theta_i}(\cdot \mid h_i)$
13:         Execute action $a_i \sim \pi_i$
14:         Sleep for duration $\Delta t_{\text{control}}$
15:     **until** Stop condition is reached
16: **end for**

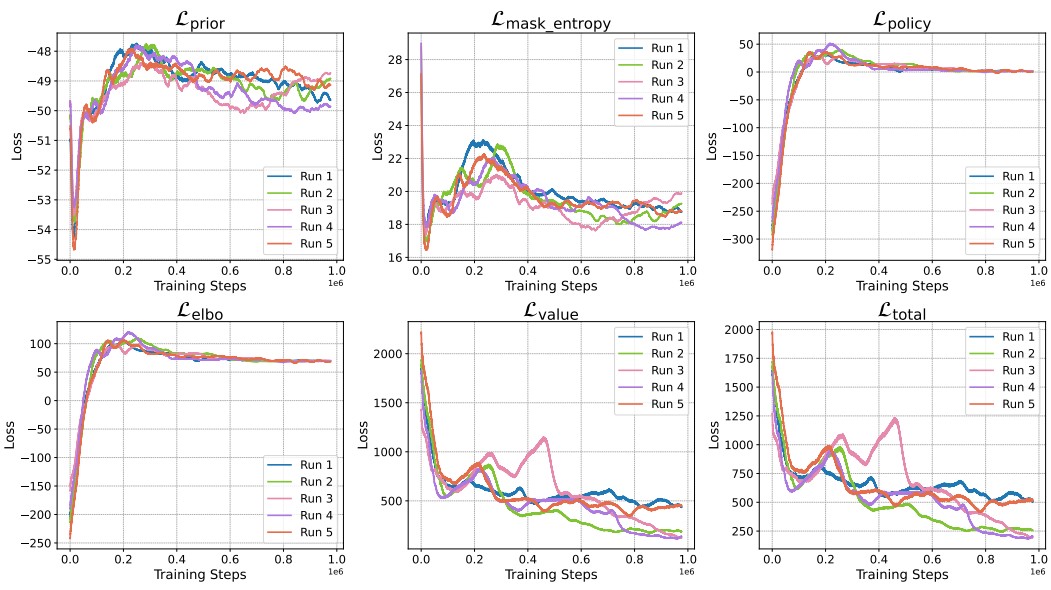

Figure 7: Training loss curves of BayesG on `ATSC_Grid`.

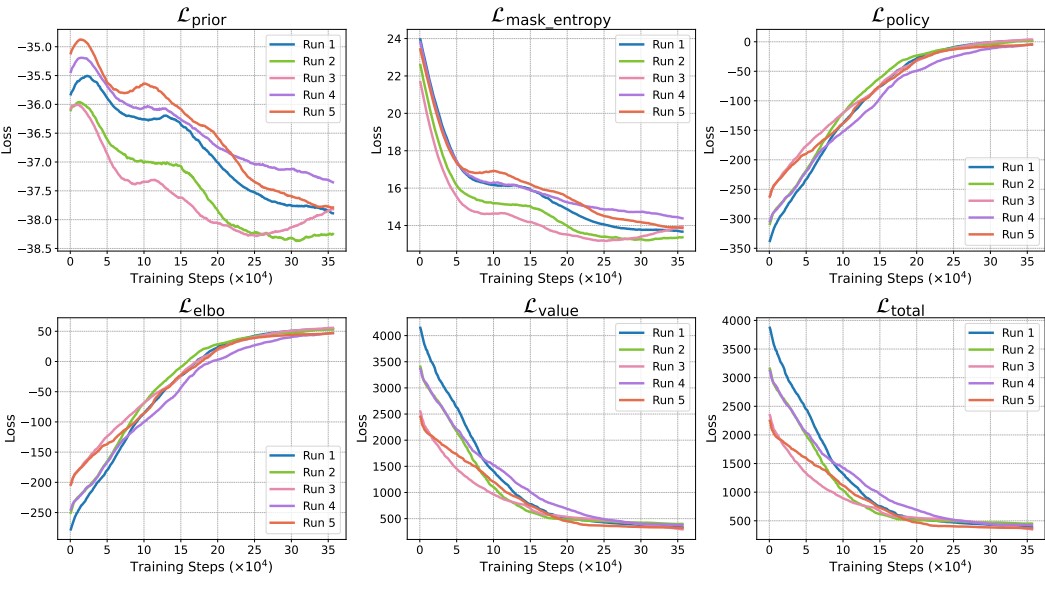

Figure 8: Training loss curves of BayesG on `Monaco`.

## E  Training Loss Analysis

Figures 7, 8, and 9 illustrate the evolution of training losses for BayesG on three representative environments: `ATSC_Grid`, `Monaco`, and `NewYork33`. We report the component-wise losses across five random seeds.

**Policy loss $\mathcal{L}_{\theta,\varphi}$.**  The policy loss reflects the negative log-probability of selected actions, weighted by the estimated advantage (see Definition 4). During training, temporary increases in this loss may occur due to distributional shift in the graph-conditioned state as the variational mask distribution $q(Z_i; \phi_i)$ evolves.  As the latent subgraphs adapt, agents may explore new action patterns that

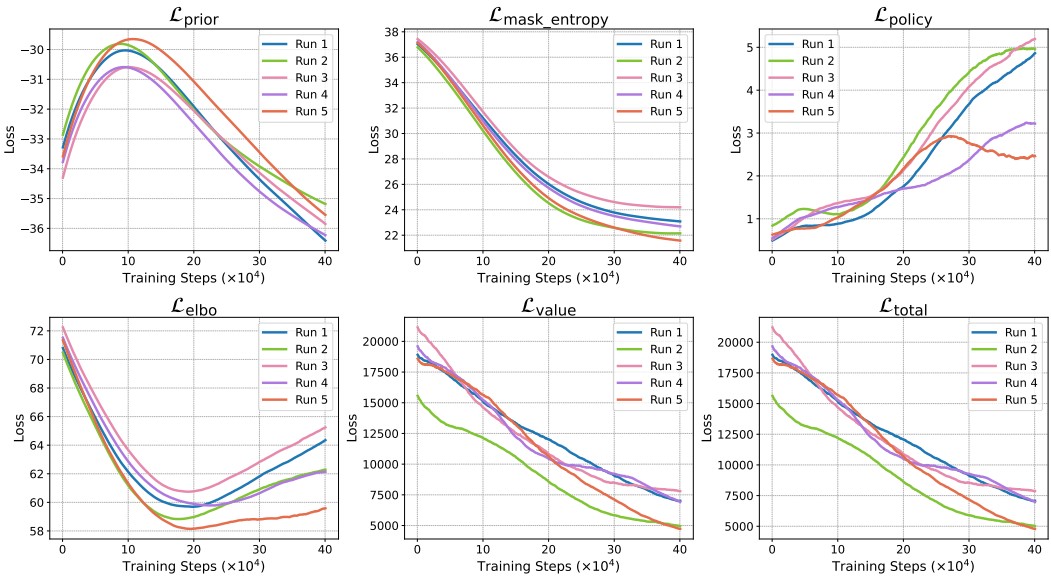

Figure 9: Training loss curves of BayesG on `NewYork33`.

momentarily reduce alignment with advantage estimates, leading to transient spikes. However, as both the policy and graph inference converge, the loss gradually stabilizes, indicating improved policy learning under the inferred interaction structures.

**ELBO loss $\mathcal{L}_{\text{ELBO}}$.** The ELBO combines the actor loss with KL regularization terms (see Definition 5). On `Grid` and `Monaco`, we observe an increasing trend in the ELBO, which stems from the increasing policy confidence (lower entropy) and the corresponding rise in the KL penalty due to deviation from the Bernoulli prior. Notably, the ELBO remains smooth and stable across seeds.

On `NewYork33`, a different pattern emerges: the ELBO decreases in early training as the agent learns useful sparse subgraphs and aligns with the prior. Later, however, as the policy sharpens and latent graphs become more deterministic, the KL term increases, causing the ELBO to rise again. This U-shaped behavior highlights a natural trade-off between policy certainty and exploration through stochastic subgraph sampling.

**Other losses.** The mask entropy term gradually decreases, indicating a transition from exploratory communication structures to more deterministic ones. The prior loss stabilizes, confirming convergence toward a learned sparsity level. The value loss steadily decreases, showing reliable value function learning.

These results confirm that BayesG effectively balances exploration and exploitation during latent graph learning and demonstrates stable convergence behavior across varying scales of networked environments.

## F Additional Visualizations on Grid Environment

To complement the qualitative analysis in Figure 3 of the main paper, we provide additional visualizations of traffic conditions and coordination patterns for **IA2C**, **CommNet**, **NeurComm**, and **BayesG** on the `ATSC_Grid` map. These snapshots are captured at multiple simulation times: 1000, 1500, 2000, 3000, and 3599 seconds.

Figure 10 displays the evolving traffic state and learned communication structure for each method. In these visualizations, the thickness and color intensity of each road segment reflect real-time vehicle density—thicker and redder segments indicate higher congestion levels.

Key observations:

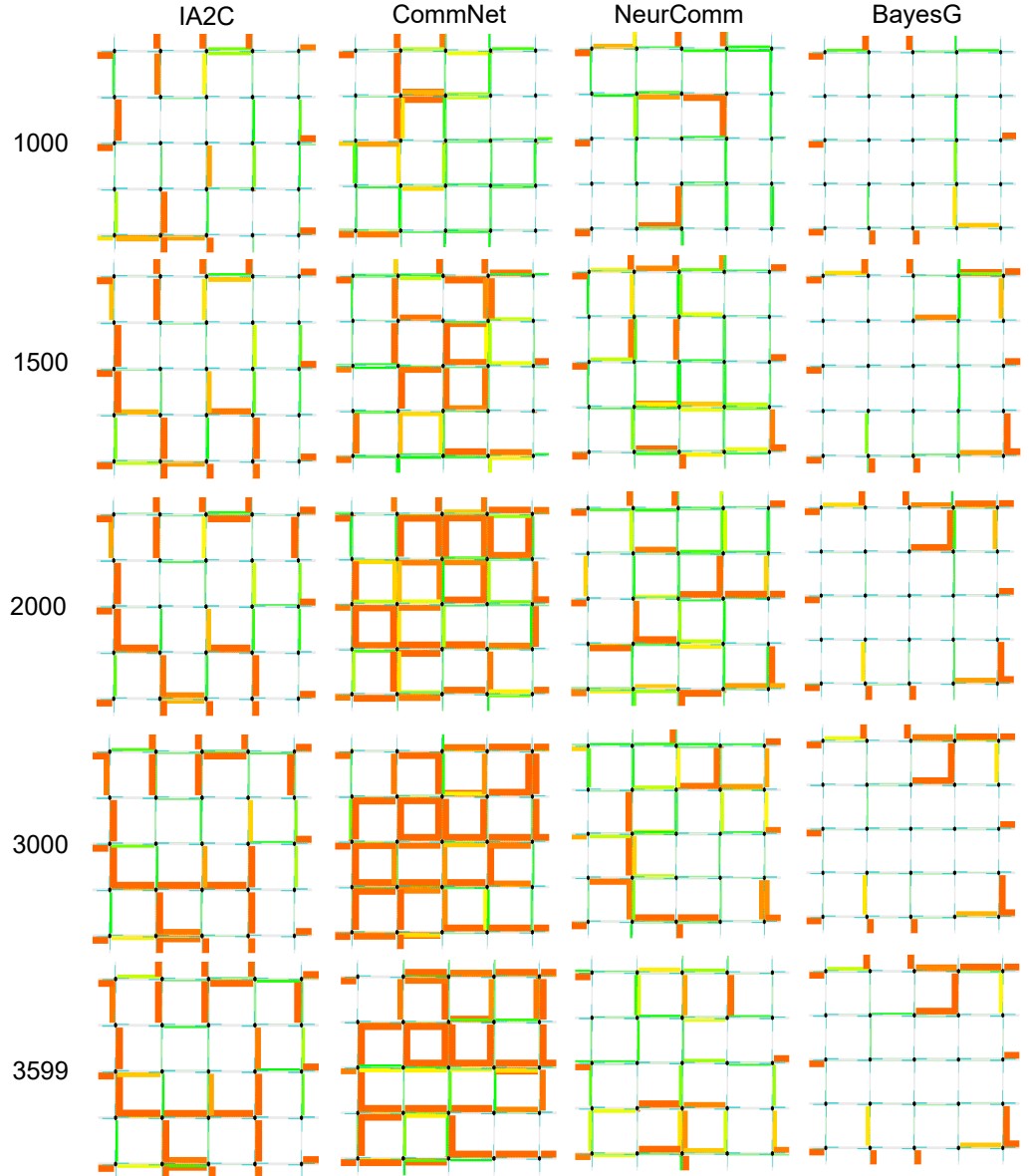

Figure 10: Qualitative comparison at different simulation times (1000–3599s) on `ATSC_Grid`. Road thickness and color represent vehicle density (thicker and redder indicates more congestion). BayesG consistently learns to avoid congestion by adaptively shaping communication, in contrast to fixed or overly dense schemes used by baselines.

- **IA2C** struggles consistently across all time points due to the absence of inter-agent communication, resulting in persistent congestion.

- **CommNet** shows slightly improved flow but lacks adaptivity, often over-communicating with irrelevant neighbors.

- **NeurComm** mitigates congestion more effectively by selectively encoding neighbor messages, though its structure remains relatively fixed.

- **BayesG** consistently exhibits the most adaptive behavior, dynamically adjusting its latent communication graph over time to emphasize critical neighbors and suppress redundant links, leading to the smoothest overall flow.

These time-resolved results further highlight the importance of dynamic, ego-graph-based communication in complex, real-world coordination tasks.

## G  Extended Case Study: Dynamic Graph Adaptation Across Time and Runs

To further validate BayesG's ability to infer dynamic and context-aware communication structures, we present additional case studies across two independent training runs on the `ATSC_Grid` environment (see Figure 11).

Each row in Figure 11 corresponds to a different simulation time step and training run: Run-1 shows the system behavior at step 600 and 1600, while Run-2 presents steps 1000 and 1400. For each time step, we visualize:

- **(a) Left:** The aggregated latent interaction matrix, where each row corresponds to an agent's inferred *ego-graph*—a probabilistic mask over its physical neighborhood. The overall matrix reflects the likelihood of communication between all agent pairs.
- **(a) Right:** The corresponding spatial network visualization of the inferred graph, where edge width encodes communication probability. This provides an interpretable view of where information is likely to flow across the physical grid.
- **(b):** The corresponding traffic density snapshot from the SUMO simulator. Redder and thicker lines denote higher congestion levels on road segments.

These results highlight how BayesG dynamically adjusts interaction structures based on evolving traffic conditions. In both runs, we observe that as local congestion increases, nearby agents increase their communication probabilities—effectively directing more coordination toward problematic areas. This behavior illustrates the emergence of adaptive, directional cooperation: upstream or adjacent intersections proactively modulate traffic signals to help alleviate pressure downstream. Such behavior aligns with human-designed traffic heuristics, where context-sensitive cooperation is essential for preventing cascading congestion.

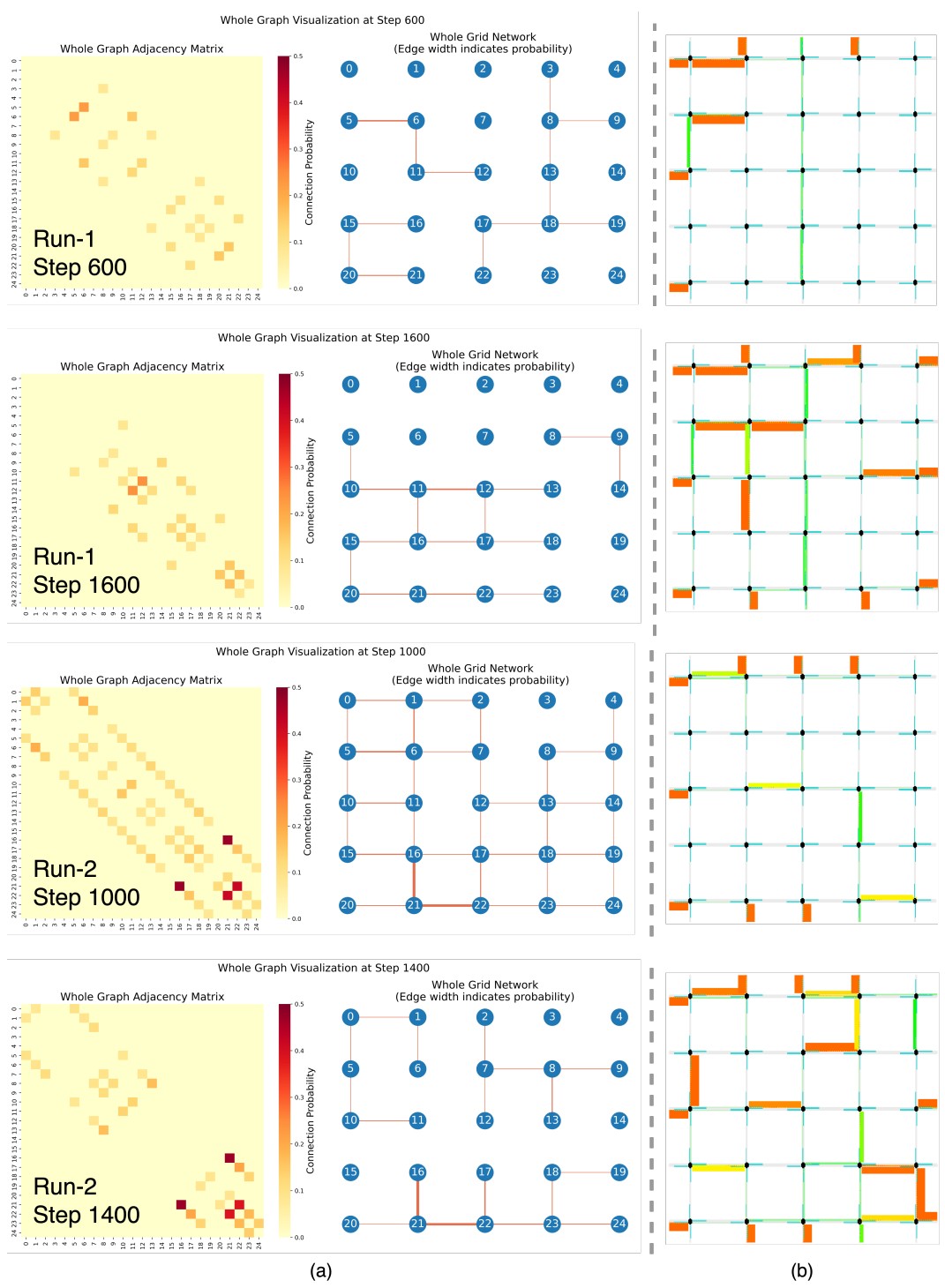

Figure 11: Extended case study visualizations for BayesG on `ATSC_Grid`. Each row shows a different simulation step and training run. **(a) Left:** Latent interaction matrix (ego-graph probabilities). **(a) Right:** Physical network view with edge thickness indicating communication strength.

Table 2: Comparison of sampling strategies in networked MARL. Our method learns context-dependent, per-edge distributions optimized jointly with the policy, in contrast to predefined or uniform sampling approaches.

| Aspect | Lin et al. [23] | Anand & Qu [27] | BayesG (Ours) |
|---|---|---|---|
| **Distribution for sampling** | Exogenous, fixed distribution $\mathcal{D}$ over active link sets $(L_t^s, L_t^r)$, sampled i.i.d.; assumes distance truncation/decay | Fixed, uniform distribution over size-$k$ subsets of local agents for value estimation and action | Learned, per-agent variational posterior $q(Z_i; \phi_i)$ over 1-hop edges; state/trajectory-dependent, trained end-to-end (ELBO) |
| **Sampling target** | Global edges that are "active" for transitions $(L^s)$ and rewards $(L^r)$ at each step | Agents (a subset of locals) for the single global agent to aggregate over (not edges) | Binary edge mask on each agent's 1-hop ego-graph (which neighbors to "listen to" for policy/communication) |
| **Locus of dynamic topology** | Dynamics-level stochasticity: new active link sets each step; multi-hop influence via chains across time | Agent-level subsampling: the global agent's view changes as different locals are sampled; topology effectively star-like, not edge-level | Policy/representation-level adaptivity: each agent samples a task-adaptive 1-hop subgraph each step; no multi-hop rewiring (multi-hop only via repeated local passes) |
| **1-hop inclusion probability** | Inclusion of $(j \rightarrow i)$ is the marginal of $\mathcal{D}$ (fixed/model-assumed; not learned or state-adaptive) | No edge-level 1-hop probability; agent-level inclusion is $k/n$ (uniform, fixed) per step for the global agent | For neighbor $j$ of agent $i$, inclusion $\Pr[z_{ij}=1] = \sigma(\phi_{ij}(\cdot))$ is learned and context-dependent (from $q$) |

# H  Detailed Comparison with Sampling-based Networked MARL

Table 2 provides a detailed side-by-side comparison of our approach with recent sampling-based methods for networked MARL [23, 27]. The comparison highlights key differences in sampling targets, distribution types, locus of adaptivity, and first-hop inclusion probabilities.

These distinctions highlight that while all three methods employ sampling for scalability, they differ fundamentally in whether the sampling distribution is learned, what structural elements are sampled, and where adaptivity occurs in the system architecture.

# I  Limitations

While BayesG demonstrates strong empirical performance and scalability on traffic signal control benchmarks, several limitations remain:

- **Fixed physical topology.** Our approach assumes a predefined, static environment graph that restricts the set of possible neighbor interactions. While this models many real-world systems such as traffic networks, it may not generalize to domains with dynamic or learned topology.

- **Local observability.** Each agent infers its latent communication graph based solely on local observations within its ego-graph. This limits the model's ability to reason over long-range dependencies that may require more global context.

- **Hyperparameter sensitivity.** The performance of the learned mask depends on the Gumbel-softmax temperature and sparsity prior. Improper tuning may lead to either under- or over-pruning of edges, potentially degrading coordination.

- **Computational cost.** Compared to non-communicative methods, BayesG introduces additional overhead due to variational sampling and masked GNN computation. While still efficient in practice, training cost may increase with neighborhood size or message-passing depth.

We believe these limitations open avenues for future work on dynamic graph adaptation, broader task domains, and further efficiency improvements.

## NeurIPS Paper Checklist

The checklist is designed to encourage best practices for responsible machine learning research, addressing issues of reproducibility, transparency, research ethics, and societal impact. Do not remove the checklist: **The papers not including the checklist will be desk rejected.** The checklist should follow the references and follow the (optional) supplemental material. The checklist does NOT count towards the page limit.

Please read the checklist guidelines carefully for information on how to answer these questions. For each question in the checklist:

- You should answer [Yes] , [No] , or [NA] .
- [NA] means either that the question is Not Applicable for that particular paper or the relevant information is Not Available.
- Please provide a short (1–2 sentence) justification right after your answer (even for NA).

**The checklist answers are an integral part of your paper submission.** They are visible to the reviewers, area chairs, senior area chairs, and ethics reviewers. You will be asked to also include it (after eventual revisions) with the final version of your paper, and its final version will be published with the paper.

The reviewers of your paper will be asked to use the checklist as one of the factors in their evaluation. While "[Yes] " is generally preferable to "[No] ", it is perfectly acceptable to answer "[No] " provided a proper justification is given (e.g., "error bars are not reported because it would be too computationally expensive" or "we were unable to find the license for the dataset we used"). In general, answering "[No] " or "[NA] " is not grounds for rejection. While the questions are phrased in a binary way, we acknowledge that the true answer is often more nuanced, so please just use your best judgment and write a justification to elaborate. All supporting evidence can appear either in the main paper or the supplemental material, provided in appendix. If you answer [Yes] to a question, in the justification please point to the section(s) where related material for the question can be found.

IMPORTANT, please:

- **Delete this instruction block, but keep the section heading "NeurIPS Paper Checklist",**
- **Keep the checklist subsection headings, questions/answers and guidelines below.**
- **Do not modify the questions and only use the provided macros for your answers**.

1. **Claims**

   Question: Do the main claims made in the abstract and introduction accurately reflect the paper's contributions and scope?

   Answer: [Yes]

   Justification: The abstract and introduction clearly describe the contributions and scope of the paper. We propose a decentralized actor–critic framework, BayesG, that learns sparse, context-aware interaction graphs over ego-graphs via Bayesian inference. The main claims regarding the formulation of a stochastic graph-based policy, integration with variational inference, and empirical performance on large-scale adaptive traffic control tasks are supported by theoretical formulation and extensive experimental results. Assumptions (e.g., fixed physical topology, decentralized observability) and limitations are appropriately stated. The results are presented in a way that reflects both their practical relevance and generalizability to other networked MARL scenarios.

   Guidelines:
   - The answer NA means that the abstract and introduction do not include the claims made in the paper.
   - The abstract and/or introduction should clearly state the claims made, including the contributions made in the paper and important assumptions and limitations. A No or NA answer to this question will not be perceived well by the reviewers.
   - The claims made should match theoretical and experimental results, and reflect how much the results can be expected to generalize to other settings.

- It is fine to include aspirational goals as motivation as long as it is clear that these goals are not attained by the paper.

2. **Limitations**

Question: Does the paper discuss the limitations of the work performed by the authors?

Answer: [Yes]

Justification: We include a dedicated "Limitations" section in the appendix, where we discuss several aspects of our approach, including assumptions on fixed topology, potential sensitivity to Gumbel-softmax temperature, and the applicability of our method to other non-graph-constrained MARL settings. We also acknowledge that our empirical validation is focused on traffic control benchmarks, and further testing in other networked domains (e.g., swarm robotics, power grids) remains future work.

Guidelines:

- The answer NA means that the paper has no limitation while the answer No means that the paper has limitations, but those are not discussed in the paper.
- The authors are encouraged to create a separate "Limitations" section in their paper.
- The paper should point out any strong assumptions and how robust the results are to violations of these assumptions (e.g., independence assumptions, noiseless settings, model well-specification, asymptotic approximations only holding locally). The authors should reflect on how these assumptions might be violated in practice and what the implications would be.
- The authors should reflect on the scope of the claims made, e.g., if the approach was only tested on a few datasets or with a few runs. In general, empirical results often depend on implicit assumptions, which should be articulated.
- The authors should reflect on the factors that influence the performance of the approach. For example, a facial recognition algorithm may perform poorly when image resolution is low or images are taken in low lighting. Or a speech-to-text system might not be used reliably to provide closed captions for online lectures because it fails to handle technical jargon.
- The authors should discuss the computational efficiency of the proposed algorithms and how they scale with dataset size.
- If applicable, the authors should discuss possible limitations of their approach to address problems of privacy and fairness.
- While the authors might fear that complete honesty about limitations might be used by reviewers as grounds for rejection, a worse outcome might be that reviewers discover limitations that aren't acknowledged in the paper. The authors should use their best judgment and recognize that individual actions in favor of transparency play an important role in developing norms that preserve the integrity of the community. Reviewers will be specifically instructed to not penalize honesty concerning limitations.

3. **Theory assumptions and proofs**

Question: For each theoretical result, does the paper provide the full set of assumptions and a complete (and correct) proof?

Answer: [NA]

Justification: The paper does not present formal theorems or propositions that require full theoretical proofs. Instead, it provides well-defined mathematical formulations (e.g., the graph-based policy, actor–critic objectives, and ELBO) grounded in variational inference and reinforcement learning. These are derived with standard assumptions and are supported by detailed derivations in Appendix A.1 and Appendix A.3. While these formulations are essential to the proposed method, they are not presented as novel theoretical results in the form of formal theorems requiring assumptions and proof structures.

Guidelines:

- The answer NA means that the paper does not include theoretical results.
- All the theorems, formulas, and proofs in the paper should be numbered and cross-referenced.

- All assumptions should be clearly stated or referenced in the statement of any theorems.
- The proofs can either appear in the main paper or the supplemental material, but if they appear in the supplemental material, the authors are encouraged to provide a short proof sketch to provide intuition.
- Inversely, any informal proof provided in the core of the paper should be complemented by formal proofs provided in appendix or supplemental material.
- Theorems and Lemmas that the proof relies upon should be properly referenced.

4. **Experimental result reproducibility**

   Question: Does the paper fully disclose all the information needed to reproduce the main experimental results of the paper to the extent that it affects the main claims and/or conclusions of the paper (regardless of whether the code and data are provided or not)?

   Answer: [Yes]

   Justification: The paper provides detailed descriptions of the experimental setup, including environment configurations (e.g., map layouts, control intervals), reward definitions, state/action representations, and the architectures used for both policy and critic networks. Implementation details for all baselines are unified under the same A2C backbone and described clearly in Section 5.1.2. Hyperparameters such as training steps, batch size, and random seed settings are also specified. Moreover, we include full algorithm pseudocode in Appendix D and derivations in Appendices A.1–A.3, which provide further clarity for reproducing our method.

   Guidelines:
   - The answer NA means that the paper does not include experiments.
   - If the paper includes experiments, a No answer to this question will not be perceived well by the reviewers: Making the paper reproducible is important, regardless of whether the code and data are provided or not.
   - If the contribution is a dataset and/or model, the authors should describe the steps taken to make their results reproducible or verifiable.
   - Depending on the contribution, reproducibility can be accomplished in various ways. For example, if the contribution is a novel architecture, describing the architecture fully might suffice, or if the contribution is a specific model and empirical evaluation, it may be necessary to either make it possible for others to replicate the model with the same dataset, or provide access to the model. In general. releasing code and data is often one good way to accomplish this, but reproducibility can also be provided via detailed instructions for how to replicate the results, access to a hosted model (e.g., in the case of a large language model), releasing of a model checkpoint, or other means that are appropriate to the research performed.
   - While NeurIPS does not require releasing code, the conference does require all submissions to provide some reasonable avenue for reproducibility, which may depend on the nature of the contribution. For example
     (a) If the contribution is primarily a new algorithm, the paper should make it clear how to reproduce that algorithm.
     (b) If the contribution is primarily a new model architecture, the paper should describe the architecture clearly and fully.
     (c) If the contribution is a new model (e.g., a large language model), then there should either be a way to access this model for reproducing the results or a way to reproduce the model (e.g., with an open-source dataset or instructions for how to construct the dataset).
     (d) We recognize that reproducibility may be tricky in some cases, in which case authors are welcome to describe the particular way they provide for reproducibility. In the case of closed-source models, it may be that access to the model is limited in some way (e.g., to registered users), but it should be possible for other researchers to have some path to reproducing or verifying the results.

5. **Open access to data and code**

   Question: Does the paper provide open access to the data and code, with sufficient instructions to faithfully reproduce the main experimental results, as described in supplemental material?

Answer: [Yes]

Justification: We provide full source code and configuration files for all experiments in the supplementary material, including detailed instructions for installation, environment setup, and execution. The traffic control benchmarks are built on SUMO, and we include the necessary network files, route generation scripts, and map preprocessing tools to reconstruct the environments. Hyperparameters and implementation details are documented to ensure faithful replication of results. All code and data will also be released publicly upon publication.

Guidelines:

- The answer NA means that paper does not include experiments requiring code.
- Please see the NeurIPS code and data submission guidelines (`https://nips.cc/public/guides/CodeSubmissionPolicy`) for more details.
- While we encourage the release of code and data, we understand that this might not be possible, so "No" is an acceptable answer. Papers cannot be rejected simply for not including code, unless this is central to the contribution (e.g., for a new open-source benchmark).
- The instructions should contain the exact command and environment needed to run to reproduce the results. See the NeurIPS code and data submission guidelines (`https://nips.cc/public/guides/CodeSubmissionPolicy`) for more details.
- The authors should provide instructions on data access and preparation, including how to access the raw data, preprocessed data, intermediate data, and generated data, etc.
- The authors should provide scripts to reproduce all experimental results for the new proposed method and baselines. If only a subset of experiments are reproducible, they should state which ones are omitted from the script and why.
- At submission time, to preserve anonymity, the authors should release anonymized versions (if applicable).
- Providing as much information as possible in supplemental material (appended to the paper) is recommended, but including URLs to data and code is permitted.

6. **Experimental setting/details**

Question: Does the paper specify all the training and test details (e.g., data splits, hyperparameters, how they were chosen, type of optimizer, etc.) necessary to understand the results?

Answer: [Yes]

Justification: The paper provides comprehensive implementation details in Section 5.1.2, including the number of training timesteps, control intervals, input feature construction, network architectures, and the use of consistent settings across baselines. Hyperparameters such as learning rates, rollout horizon, optimizer type (RMSprop), and entropy coefficients are documented. Additionally, environment-specific simulation details—such as episode lengths, control intervals, and traffic statistics—are described in the Experimental Setup section. Further details are included in the supplementary material to ensure transparency and reproducibility.

Guidelines:

- The answer NA means that the paper does not include experiments.
- The experimental setting should be presented in the core of the paper to a level of detail that is necessary to appreciate the results and make sense of them.
- The full details can be provided either with the code, in appendix, or as supplemental material.

7. **Experiment statistical significance**

Question: Does the paper report error bars suitably and correctly defined or other appropriate information about the statistical significance of the experiments?

Answer: [Yes]

Justification: The experimental results report mean episode returns across 5 random seeds, with shaded error regions representing one standard deviation. This provides a clear visualization of the variance and stability of different methods. All performance curves are smoothed using a moving average for readability while preserving variability trends. These practices follow common standards for statistical reporting in reinforcement learning research.

Guidelines:

- The answer NA means that the paper does not include experiments.
- The authors should answer "Yes" if the results are accompanied by error bars, confidence intervals, or statistical significance tests, at least for the experiments that support the main claims of the paper.
- The factors of variability that the error bars are capturing should be clearly stated (for example, train/test split, initialization, random drawing of some parameter, or overall run with given experimental conditions).
- The method for calculating the error bars should be explained (closed form formula, call to a library function, bootstrap, etc.)
- The assumptions made should be given (e.g., Normally distributed errors).
- It should be clear whether the error bar is the standard deviation or the standard error of the mean.
- It is OK to report 1-sigma error bars, but one should state it. The authors should preferably report a 2-sigma error bar than state that they have a 96% CI, if the hypothesis of Normality of errors is not verified.
- For asymmetric distributions, the authors should be careful not to show in tables or figures symmetric error bars that would yield results that are out of range (e.g. negative error rates).
- If error bars are reported in tables or plots, The authors should explain in the text how they were calculated and reference the corresponding figures or tables in the text.

8. **Experiments compute resources**

Question: Does the paper report error bars suitably and correctly defined or other appropriate information about the statistical significance of the experiments?

Answer: [Yes]

Justification: The experimental results report mean episode returns across 5 random seeds, with shaded error regions representing one standard deviation. This provides a clear visualization of the variance and stability of different methods. All performance curves are smoothed using a moving average for readability while preserving variability trends. These practices follow common standards for statistical reporting in reinforcement learning research.

Guidelines:

- The answer NA means that the paper does not include experiments.
- The paper should indicate the type of compute workers CPU or GPU, internal cluster, or cloud provider, including relevant memory and storage.
- The paper should provide the amount of compute required for each of the individual experimental runs as well as estimate the total compute.
- The paper should disclose whether the full research project required more compute than the experiments reported in the paper (e.g., preliminary or failed experiments that didn't make it into the paper).

9. **Code of ethics**

Question: Does the research conducted in the paper conform, in every respect, with the NeurIPS Code of Ethics https://neurips.cc/public/EthicsGuidelines?

Answer: [Yes]

Justification: The research does not involve human subjects, sensitive data, or deployable systems. It focuses solely on algorithmic development and empirical evaluation in simulated traffic control environments. All experiments were conducted in accordance with NeurIPS ethical guidelines, with no foreseeable risk of harm, privacy concerns, or misuse.

Guidelines:

- The answer NA means that the authors have not reviewed the NeurIPS Code of Ethics.
- If the authors answer No, they should explain the special circumstances that require a deviation from the Code of Ethics.
- The authors should make sure to preserve anonymity (e.g., if there is a special consideration due to laws or regulations in their jurisdiction).

10. **Broader impacts**

Question: Does the paper discuss both potential positive societal impacts and negative societal impacts of the work performed?

Answer: [Yes]

Justification: The paper includes a discussion of societal impacts in the Limitations section. On the positive side, BayesG has the potential to improve urban traffic efficiency, reduce congestion, and lower carbon emissions. On the negative side, if deployed unfairly, such systems might unintentionally prioritize certain routes or neighborhoods, raising fairness concerns. Additionally, reliance on traffic data may introduce biases if the input data is unbalanced or misrepresentative.

Guidelines:

- The answer NA means that there is no societal impact of the work performed.
- If the authors answer NA or No, they should explain why their work has no societal impact or why the paper does not address societal impact.
- Examples of negative societal impacts include potential malicious or unintended uses (e.g., disinformation, generating fake profiles, surveillance), fairness considerations (e.g., deployment of technologies that could make decisions that unfairly impact specific groups), privacy considerations, and security considerations.
- The conference expects that many papers will be foundational research and not tied to particular applications, let alone deployments. However, if there is a direct path to any negative applications, the authors should point it out. For example, it is legitimate to point out that an improvement in the quality of generative models could be used to generate deepfakes for disinformation. On the other hand, it is not needed to point out that a generic algorithm for optimizing neural networks could enable people to train models that generate Deepfakes faster.
- The authors should consider possible harms that could arise when the technology is being used as intended and functioning correctly, harms that could arise when the technology is being used as intended but gives incorrect results, and harms following from (intentional or unintentional) misuse of the technology.
- If there are negative societal impacts, the authors could also discuss possible mitigation strategies (e.g., gated release of models, providing defenses in addition to attacks, mechanisms for monitoring misuse, mechanisms to monitor how a system learns from feedback over time, improving the efficiency and accessibility of ML).

11. **Safeguards**

Question: Does the paper describe safeguards that have been put in place for responsible release of data or models that have a high risk for misuse (e.g., pretrained language models, image generators, or scraped datasets)?

Answer: [NA]

Justification: The proposed method, BayesG, is designed for decentralized coordination in networked multi-agent systems, particularly adaptive traffic signal control. It does not involve the release of models or data with high risk for misuse, such as generative models or large-scale pretrained models. Therefore, this question is not applicable.

Guidelines:

- The answer NA means that the paper poses no such risks.
- Released models that have a high risk for misuse or dual-use should be released with necessary safeguards to allow for controlled use of the model, for example by requiring that users adhere to usage guidelines or restrictions to access the model or implementing safety filters.

- Datasets that have been scraped from the Internet could pose safety risks. The authors should describe how they avoided releasing unsafe images.
- We recognize that providing effective safeguards is challenging, and many papers do not require this, but we encourage authors to take this into account and make a best faith effort.

12. **Licenses for existing assets**

Question: Are the creators or original owners of assets (e.g., code, data, models), used in the paper, properly credited and are the license and terms of use explicitly mentioned and properly respected?

Answer: [Yes]

Justification: All third-party assets used in the paper, including simulation environments (e.g., SUMO), baseline implementations, and datasets, are properly cited in the main text. The terms of use and licenses for these assets (e.g., SUMO under Eclipse Public License) have been reviewed and respected. Any reused code or models are referenced appropriately in the implementation section.

Guidelines:

- The answer NA means that the paper does not use existing assets.
- The authors should cite the original paper that produced the code package or dataset.
- The authors should state which version of the asset is used and, if possible, include a URL.
- The name of the license (e.g., CC-BY 4.0) should be included for each asset.
- For scraped data from a particular source (e.g., website), the copyright and terms of service of that source should be provided.
- If assets are released, the license, copyright information, and terms of use in the package should be provided. For popular datasets, `paperswithcode.com/datasets` has curated licenses for some datasets. Their licensing guide can help determine the license of a dataset.
- For existing datasets that are re-packaged, both the original license and the license of the derived asset (if it has changed) should be provided.
- If this information is not available online, the authors are encouraged to reach out to the asset's creators.

13. **New assets**

Question: Are new assets introduced in the paper well documented and is the documentation provided alongside the assets?

Answer: [Yes]

Justification: The paper introduces a new decentralized MARL framework (BayesG) and associated implementation. Full documentation, including code structure, environment setup, and reproducibility instructions, is provided in the supplementary material and will be made publicly available upon publication.

Guidelines:

- The answer NA means that the paper does not release new assets.
- Researchers should communicate the details of the dataset/code/model as part of their submissions via structured templates. This includes details about training, license, limitations, etc.
- The paper should discuss whether and how consent was obtained from people whose asset is used.
- At submission time, remember to anonymize your assets (if applicable). You can either create an anonymized URL or include an anonymized zip file.

14. **Crowdsourcing and research with human subjects**

Question: For crowdsourcing experiments and research with human subjects, does the paper include the full text of instructions given to participants and screenshots, if applicable, as well as details about compensation (if any)?

Answer: [NA]

Justification: This paper does not involve any crowdsourcing experiments or human subject research.

Guidelines:

- The answer NA means that the paper does not involve crowdsourcing nor research with human subjects.
- Including this information in the supplemental material is fine, but if the main contribution of the paper involves human subjects, then as much detail as possible should be included in the main paper.
- According to the NeurIPS Code of Ethics, workers involved in data collection, curation, or other labor should be paid at least the minimum wage in the country of the data collector.

15. **Institutional review board (IRB) approvals or equivalent for research with human subjects**

Question: Does the paper describe potential risks incurred by study participants, whether such risks were disclosed to the subjects, and whether Institutional Review Board (IRB) approvals (or an equivalent approval/review based on the requirements of your country or institution) were obtained?

Answer: [NA]

Justification: This paper does not involve human subjects or participant-based studies and therefore does not require IRB approval or related disclosures.

Guidelines:

- The answer NA means that the paper does not involve crowdsourcing nor research with human subjects.
- Depending on the country in which research is conducted, IRB approval (or equivalent) may be required for any human subjects research. If you obtained IRB approval, you should clearly state this in the paper.
- We recognize that the procedures for this may vary significantly between institutions and locations, and we expect authors to adhere to the NeurIPS Code of Ethics and the guidelines for their institution.
- For initial submissions, do not include any information that would break anonymity (if applicable), such as the institution conducting the review.

16. **Declaration of LLM usage**

Question: Does the paper describe the usage of LLMs if it is an important, original, or non-standard component of the core methods in this research?

Answer: [NA]

Justification: This paper does not incorporate large language models (LLMs) as part of its core methods. Any usage of LLMs was limited to minor writing support and did not affect the scientific contributions or methodology of the work.

Guidelines:

- The answer NA means that the core method development in this research does not involve LLMs as any important, original, or non-standard components.
- Please refer to our LLM policy (https://neurips.cc/Conferences/2025/LLM) for what should or should not be described.

