# OpenReview forum: "Bayesian Ego-graph Inference for Networked Multi-Agent Reinforcement Learning"
_NeurIPS.cc/2025/Conference — NeurIPS 2025 poster_

### Official Review · Reviewer_GrDU · 2025-06-22

**Clarity:** 2
**Significance:** 4
**Originality:** 3
**Rating:** 4
**Confidence:** 4

**Summary:**

This paper presents BayesG, a novel variational Bayesian inference framework designed for networked multi-agent reinforcement learning (Networked-MARL). BayesG enables each agent to make decisions based on a sampled subgraph of its local physical neighborhood, thereby achieving sparse and context-aware interaction structures. The method integrates variational Bayesian inference with A2C, employing the evidence lower bound (ELBO) objective function for joint optimization. This allows for dynamic adjustment of the interaction structures among agents, thereby reducing unnecessary communication overheads. Experiments conducted in five adaptive traffic signal control (ATSC) benchmark scenarios demonstrate that BayesG outperforms existing MARL baselines in terms of performance, convergence speed, and scalability, highlighting its potential and applicability in complex networked systems.

**Questions:**

Please refer to the section "Weaknesses" for detailed information.

**Ethical Concerns:**

["NO or VERY MINOR ethics concerns only"]

**Final Justification:**

**Final Decision**: I have decided to raise my score to 4.

**Justification**: The paper mainly considers BayesG, enables each agent to make decisions based on a sampled subgraph of its local physical neighborhood, thereby achieving sparse and context-aware interaction structures. Overall, the writing is clear and the motivation is relatively clear. During the rebuttal phase, the issues that have been resolved through active interaction with the authors are: clarification of the second paragraph; theoretical guarantees, discussion on non-stationarity, and distinction from related work; the issues not addressed due to content and space limitations are modifications of minor errors. Combined with relevant comments from other rebuttals, I now understand the contributions of this paper, and I have decided to raise my score to 4.

**Limitations:**

Yes.

**Paper Formatting Concerns:**

No major formatting issues in this paper.

**Quality:**

3

**Strengths And Weaknesses:**

**Strengths**
﻿
1. The study holds a certain degree of value.
2. The associated experiments are relatively comprehensive.
﻿
**Weaknesses**
1. There are quite a few grammatical and formatting errors in this paper. For instance, the legend of Fig 2 should be placed at the top rather than the bottom; there are missing punctuation marks in equation (12), etc. It is suggested that the authors review paper carefully.
﻿
2. Section 4.2 appears to be poorly written, which I find very difficult to comprehend. I would recommend expanding this section to clarify the rationale behind the use of the Bayesian approach, as well as the specific implementation details. This would enhance the readability significantly.
﻿
3. Does this work provide any theoretical guarantees? As is well-known, policy iteration under DTDE typically encounters the issue of non-stationarity, yet I have not seen how this paper addresses this problem.
﻿
4. Still regarding the issue of non-stationarity, employing a learning-based method for sampling is an intuitive idea. However, does such sampling have any impact on the stability of policy iteration?
﻿
If you are willing to provide clear answers to these questions, I would consider raising my score.

---

> ### Author Rebuttal · Authors · 2025-07-30
>
> > **Grammatical and formatting issues (e.g., Fig. 2 legend, punctuation in Eq. (12))**
>
> We appreciate this comment and will thoroughly revise the manuscript in the final version, including:
>
> - Moving all legends (e.g., Fig. 2) to consistent and appropriate positions.
>
> - Carefully proofread all equations, especially Eq. (12), to ensure correct punctuation and presentation.
>
> - Conducting a full grammatical and typographic pass.
>
> > **Clarity and completeness of Section 4.2 (Bayesian Graph Inference)**
>
> Thank you for pointing this out. We agree that Section 4.2 is central to the paper and could benefit from expanded exposition. In the final version:
>
> - We will revise this section to clarify the motivation for using Bayesian inference, specifically, to enable uncertainty-aware edge masking and regularized graph structure learning.
>
> - We will better explain how the variational distribution over binary masks is amortized using graph neural networks and trained end-to-end with the policy.
>
> - We will also clarify the interaction between the ELBO objective and policy optimization.
>
> We also note that a more detailed explanation of this mechanism and architecture is currently provided in the Appendix, but we agree that bringing some of this detail into the main text will improve accessibility and understanding.
>
>
> > **Theoretical guarantees and handling of non-stationarity in DTDE settings**
>
> We appreciate the reviewer’s insightful question regarding theoretical guarantees and non-stationarity in decentralized MARL.
>
> 1. **Non-stationarity in DTDE**
>
> In DTDE settings, each agent learns its policy independently while treating the environment as stationary. However, in multi-agent scenarios, the environment includes other agents whose policies are continuously evolving during training. This results in **non-stationary learning signals**, which often degrade stability and convergence in policy iteration. This issue is well-documented in prior works (e.g., Hernandez-Leal et al., 2019 [survey on MARL]).
>
> 2. **How CTDE alleviates non-stationarity**
>
> Centralized Training with Decentralized Execution (CTDE) mitigates this issue by providing each agent access to **global information or centralized critics** during training. This stabilizes learning by accounting for co-adaptation among agents. Prior works such as QMIX (ICML 2018) and MADDPG (ICML 2017) have shown that centralized critics can **eliminate or significantly reduce non-stationarity during training**, leading to more stable and sample-efficient learning.
>
> 3. **Our relationship to CTDE via Spatiotemporal MDP**
>
> Our work lies **between DTDE and CTDE** in terms of structural assumptions. Specifically, we operate under the Spatiotemporal MDP framework (Definition 1), in which:
>
> - Each agent's **transition** and **reward** are conditioned on its **physical neighborhood**.
>
> - This imposes a partial coupling across agents via a fixed environment graph (e.g., road topology), ensuring that agents only adapt to **localized dynamics** rather than the entire system.
>
> - Consequently, the variability perceived by each agent is constrained to its **stable local neighborhood**, substantially reducing global non-stationarity.
>
> Through this **structural prior** and **ego-centric masking**, our approach preserves decentralized execution while benefiting from the stability typically associated with CTDE. While we do not offer formal convergence guarantees, our design choices are empirically validated in terms of training stability and sample efficiency (Section 5.4, Fig. 5).
>
> **On the lack of formal analysis:**
> We also note that many recent NeurIPS publications proposing novel methods for MARL similarly do not provide theoretical guarantees, focusing instead on empirical performance and structural insights. These include:
>
> - Learning to Share in Multi-Agent Reinforcement Learning (NeurIPS 2022)
>
> - Selectively Sharing Experiences Improves Multi-Agent Reinforcement Learning (NeurIPS 2023)
>
> - Automatic Grouping for Efficient Cooperative Multi-Agent Reinforcement Learning (NeurIPS 2023)
>
> We will expand our discussion in the final version to better articulate the structural mitigation of non-stationarity via spatiotemporal MDPs and position our approach relative to CTDE and DTDE.
>
> > **Whether the learned graph sampling process impacts the stability of policy iteration**
>
> This is an insightful concern. Our response has two parts:
>
> - Learning a variational distribution over edge masks allows us to capture epistemic uncertainty over which neighbors are most informative. This stochasticity can improve robustness in dynamic or partially observable environments, which reflects real-world scenarios (e.g., noisy or fluctuating traffic patterns).
>
> - The sampled communication graph $Z_i$ influences the actor loss via the graph-conditioned policy $\tilde{\pi}_i$, as shown in Definition 4. This actor loss is embedded in the likelihood term of our ELBO formulation:
> `L_ELBO = E_{q(Z_i; φ_i)}[ -L_{θ, φ} + log p(Z_i) - log q(Z_i; φ_i) ]`.This setup ensures that mask sampling is directly driven by task performance—**graphs that yield better policies are more likely under the learned posterior.**
>
> Therefore, our sampling-based approach is not only **grounded probabilistically** but also aligned with policy improvement. Empirically, as shown in our ablation studies (e.g., Fig. 5), this coupling leads to more **stable and performant learning** than alternatives like deterministic graph construction or random masking.

---

> ### Comment · Reviewer_GrDU · 2025-08-06
>
> Thank you for your response! I still have a few concerns regarding this work:
> 1. For convergence analysis of related methods, I would like to draw the authors’ attention to [1] and [2].
> 2. Compared to [1] and [2], it might be helpful to include a discussion comparing this work with similar approximation-based approaches. I notice that in the context of networked MARL, several existing methods already employ sampling strategies—[1] and [2] being good examples. What is the key distinction between the proposed method and these prior works?
> Given time constraints, I understand that additional experiments may not be feasible. However, I encourage the authors to focus on clearly articulating how their approach differs from and advances beyond these closely related methods. Addressing this point would significantly help me better evaluate the paper’s contribution, as I currently have some unresolved concerns.
>
> **Reference**
>
> [1] Lin Y, Qu G, Huang L, et al. Multi-agent reinforcement learning in stochastic networked systems. In NeurIPS, 2021.
>
> [2] Anand E, Qu G. Efficient reinforcement learning for global decision making in the presence of local agents at scale. arXiv preprint arXiv:2403.00222, 2024.

---

> > ### Author Response · Authors · 2025-08-07
> > **Response to Reviewer: Clarifying Differences**
> >
> > Thanks for the thoughtful pointer and the concrete references. We appreciate the reviewer’s emphasis on positioning our method relative to approximation-based sampling approaches such as **[1]** Lin et al. (NeurIPS’21) and **[2]** Anand & Qu (arXiv:2403.00222). We have carefully read both, as well as the related **[3] Qu et al., NeurIPS’20** (“Scalable MARL for Networked Systems with Average Reward”). We summarise the differences between ours and theirs below:
> >
> >
> > ## How our approach differs—high-level summary
> >
> > 1) **What is sampled and why?**
> >    - **[1]** introduces a *non-learned*, fixed distribution $\mathcal{D}$ over active link sets for **dynamics/rewards**, facilitating µ-decay style analyses.
> >    - **[2]** uses *uniform subsampling of agents* (size-$k$) for a **single global decision maker**, yielding mean-field–type approximation and error bounds.
> >    - **Ours** differs in **both object and purpose of sampling**: each (decentralized) agent learns a **variational posterior over its 1-hop edges** and samples a **binary mask** that gates *policy/representation* (not the environment). The distribution is **learned, context-dependent**, and optimized via an ELBO jointly with the policy to achieve **task-adaptive sparsity**.
> >
> > 2) **Varying topology, but at different levels.**
> >    - **[1]** uses time-varying active link sets in the **environment-dynamics level**; multi-hop influence then emerges across steps.
> >    - **[2]** repeatedly **reshuffles** the subset of local agents consulted by the global controller at the **decision-making level** (star-shaped view).
> >    - **Ours** keeps the physical 1-hop graph fixed while making the **communication level** adaptive via learned, per-edge inclusion—so the variability resides in the communication mechanism rather than in the environment dynamics or agent selection.
> >
> >
> > 3) **First-hop probabilities.**
> >    - **[1]**: 1-hop inclusion is the marginal of $ \mathcal{D}$ (fixed/pre-defined; not state-adaptive).
> >    - **[2]**: inclusion is uniform $k/n$ at the **agent** level (no edge-level notion).
> >    - **Ours**: 1-hop inclusion probability is **learned** (Bernoulli per edge with reparameterization), **state/trajectory-dependent**, and optimized for control.
> >
> > 4) **Complementarity to convergence analyses.**
> >    - **[1]** and **[3]** provide convergence under **structural assumptions on the environment and value decay** (e.g., distance truncation/decay, average-reward settings), while **[2]** gives **approximation guarantees** from uniform subsampling for a global agent.
> >    - Our present contribution is **algorithmic/architectural**: we integrate the sampling distribution into the **policy/representation layer** to adaptively select informative first-hop neighbors. Our method differs fundamentally from non-learned-link and global-sampling frameworks. **Extending their theoretical tools (e.g., µ-decay analysis or mean-field approximations) to our learned 1-hop masking setup is a promising direction for future work**. We will explicitly discuss this connection and outline such extensions in the revision.
> >
> >
> > ## Where does this place our contribution
> >
> > - **Advances beyond [1], [2]:** We replace predefined or uniform sampling with a **learned, per-edge, per-agent** distribution that is **optimized jointly with the policy**. This yields **task-adaptive, topology-respecting** sparsity right where it matters for decentralized control and representation, while remaining compatible with standard 1-hop communication constraints.
> > - **Relation to [3]:** Their scalable framework for networked systems under average-reward offers structural insights and convergence under specific assumptions. Our method can be viewed as a complementary layer that **learns the effective 1-hop neighborhood** feeding the controller; we will discuss how their analytical tools might be adapted to our learned-mask setting (e.g., bounding performance as a function of mask sparsity and local signal quality).
> >
> > **Action in the final version.** We will (i) include a focused subsection contrasting the sampling distributions, sampling targets, and dynamic-topology mechanisms across **[1]**, **[2]**, and **[3]**, and (ii) expand our related-work discussion to clearly articulate how our **learned, context-adaptive 1-hop edge masks** differ from (and can complement) non-learned link activation and uniform agent subsampling approaches. We believe this clarifies the conceptual novelty and practical value of our method and directly addresses the reviewer’s concerns.
> >
> > **References**
> > [1] Lin Y, Qu G, Huang L, et al. Multi-agent reinforcement learning in stochastic networked systems. In NeurIPS, 2021.
> > [2] Anand E, Qu G. Efficient reinforcement learning for global decision making in the presence of local agents at scale. arXiv:2403.00222, 2024.
> > [3] Qu G, Lin Y, Wierman A, Li N. Scalable Multi-Agent Reinforcement Learning for Networked Systems with Average Reward. NeurIPS, 2020.

---

> > > ### Comment · Reviewer_GrDU · 2025-08-07
> > >
> > > Thank you very much for your response! My concerns have been fully addressed. The work appears interesting and fills some gaps with the existing studies I am aware of. I will raise my score accordingly. Best of luck!

---

> > > > ### Author Response · Authors · 2025-08-07
> > > >
> > > > Thank you for your valuable feedback. As you mentioned about raising the score, I’m looking forward to seeing the updated evaluation.

---

### Official Review · Reviewer_ZRqv · 2025-06-24

**Clarity:** 2
**Significance:** 3
**Originality:** 3
**Rating:** 4
**Confidence:** 2

**Summary:**

This paper studies networked multi-agent reinforcement learning (Networked-MARL), where each agent’s transition probability only depends on local neighborhood states and actions. The authors propose a new policy class for each agent that samples a subgraph over its local neighborhood. The policy decides local actions based on the information within this masked subgraph. The authors formulate the problem of learning latent edge masks as a Bayesian inference problem. As a result, the variational distribution can be trained end-to-end using an evidence lower bound objective. Intuitively, this policy class allows each agent to emphasize high-impact neighbors and suppress irrelevant ones. The authors test their methods on a traffic signal control task.

**Questions:**

In the rebuttal, I hope the authors can answer my questions from the previous section and clarify the missing content.

**Ethical Concerns:**

["NO or VERY MINOR ethics concerns only"]

**Final Justification:**

Since the authors did not provide additional ablation studies to support the intuition behind graph masking strategies, I would like to keep my score unchanged at the borderline accept.

**Limitations:**

Yes.

**Paper Formatting Concerns:**

N/A.

**Quality:**

3

**Strengths And Weaknesses:**

Strengths: It’s interesting to see that a more sophisticated design of the RL policy class can improve the performance of Networked-MARL. Emphasizing more relevant neighbors makes sense, especially when the network graph is dense, and most neighbors/information may be irrelevant. It’s also good to see that the latent edge masks can be learned end-to-end. In the ablation studies, the authors make the point that such graph masking is critical.

Weakness: Section 3 about the preliminaries is too concise. The authors should make the section self-contained rather than relying on references. I also suggest dividing this section into two subsections. The first one can focus on the MDP environment, i.e., what is special about the transition probability, rewards, and available information. The second one discusses the existing algorithm and its limitations.

Section 5.1 about the experiment setup also misses important details. Specifically, I encourage the authors to explain what the states, actions, transition probabilities, and rewards correspond to in the application of adaptive traffic signal control. And, most importantly, why does this application (approximately) satisfy Definition 1?

Since a key novelty of this work is about graph masking strategies, I think the first ablation study in Section 5.4 should be strengthened. The reader might ask when the graph masking strategies enable the most significant improvements over no masking. From Figure 5, the improvement on the dataset ‘NewYork 51’ is not as significant as ‘Monaco.’ I think the authors should do this ablation study on all ATSC environments and discuss the effectiveness of graph masking strategies. For example, does it improve more on dense networks?

---

> ### Author Rebuttal · Authors · 2025-07-30
>
> > **Conciseness of Section 3 and suggestions to restructure it.**
>
> Thank you for the suggestion. Due to space limitations in the submission version, we provided a condensed version of the background. However, we acknowledge the importance of making the preliminaries more self-contained and structured. In the final version, we will:
>
> Restructure Section 3 into two subsections:
> - One focusing on the spatiotemporal MDP setting, clearly explaining how state transitions, rewards, and local observability are defined under neighborhood-based interactions.
> - The second summarizes key prior approaches and their limitations in the context of dynamic, decentralized interaction modeling.
>
> We will also move some essential formal definitions from the appendix to the main text in the final version to enhance readability and self-containment.
>
> -----------------------
> > **Details of adaptive traffic signal control (ATSC) in Section 5.1 and justification for Definition 1**
>
> We appreciate this question and the opportunity to clarify. In Section 5.1, we briefly described the mapping from the real-world traffic environment to the MDP components. To elaborate:
>
> - **States**: Each agent observes its local traffic state via sensors such as induction loop detectors (ILDs), which provide vehicle density, queue length, and waiting times on incoming lanes.
>
> - **Actions**: The action space of each agent consists of phase control decisions (e.g., switching between green and red for traffic directions).
>
> - **Transition Probabilities**: These are not analytically available but are implicitly defined via the microscopic traffic simulator (SUMO). The transition of each agent’s local traffic state depends primarily on its own action and the actions of neighboring intersections (e.g., upstream traffic release).
>
> - **Rewards**: Defined as the (negative) number of halted vehicles at the intersection, normalized by a fixed scale. This encourages reducing traffic congestion locally.
>
> These components are naturally aligned with the **Spatiotemporal MDP** framework (Definition 1) because:
>
> - The transition dynamics are **localized and depend on neighborhood interactions**, consistent with the assumption that `p_i(s_i'| s_{V_i}, u_i, u_{N_i})`.
>
> - The reward function is **decomposed per agent**, depending only on local state-action pairs.
>
> - The neighborhood structure is explicitly defined by the **physical road network graph**, which is fixed and sparse.
>
> Moreover, **centralized training methods are often impractical or suboptimal** in ATSC due to several real-world constraints:
> - **Scalability**: As the size of the traffic network grows (e.g., in city-scale scenarios), centralized models become computationally expensive and suffer from poor generalization due to the curse of dimensionality in joint state-action spaces.
> - **Communication constraints**: Centralized training assumes full observability or global message aggregation, which is unrealistic in large distributed traffic systems with bandwidth and latency limitations.
>
> Thus, our formulation — learning decentralized policies that respect the fixed physical connectivity and local dynamics — not only satisfies the assumptions of the spatiotemporal MDP, but also reflects the operational needs of real-world traffic control systems.
>
> We will revise Section 5.1 to clarify these mappings and add this justification. We also plan to release our code and ATSC simulation environments upon publication.
>
> -----------------------
> > **Strengthening the ablation study of graph masking strategies**
>
> We agree that understanding when graph masking yields the most benefit is valuable. While our current ablation study includes two maps (Monaco and NewYork51) with different characteristics, we did not include all five due to time constraints during the rebuttal phase. We will extend this study across all ATSC maps in the final version.
>
> **Regarding the effectiveness of masking**:
>
> - According to our analysis, the benefit of graph masking is more pronounced in **denser physical networks**, where agents are connected to more neighbors. In such settings, the probability of receiving **redundant or irrelevant information increases**, which can negatively impact policy learning. Our learned masks help filter out such noisy edges, promoting more focused and effective communication.
>
> - In sparser environments, where each agent has fewer neighbors by design, the amount of incoming noise is naturally limited, and thus the marginal benefit of masking is expected to be smaller.
>
> **On the apparent difference between Monaco and NewYork51 in Figure 5:**
>
> The seemingly smaller improvement on NewYork51 is partially due to scale compression, caused by the relatively poor performance of the RandMask baseline. In absolute terms: The performance gap between BayesG and NoMask is ~100 in Monaco and ~60–70 in NewYork51, both of which are substantial.
>
> We will clarify this interpretation in the revised version and present both raw and normalized performance gains in the final plots.

---

> > ### Comment · Reviewer_ZRqv · 2025-08-05
> >
> > Thank you for confirming that you agree with most of the suggestions. The intuitive explanation of the graph masking strategies makes sense, and I believe it's critical to provide sufficient supporting evidence in the final version. I will leave my score unchanged.

---

### Official Review · Reviewer_2Jg6 · 2025-07-06

**Clarity:** 3
**Significance:** 2
**Originality:** 2
**Rating:** 4
**Confidence:** 4

**Summary:**

This paper introduces BayesG, a MARL method for cases when an individual agent can pass messages to others in its neighborhood. The novel technique at the core of BayesG is the use of Bayesian variational inference to learn sparse, context-aware interaction graphs. Each agent operates on a localized ego-graph and learns to sample a latent binary mask over its neighbors. This mask dynamically prunes communication links, allowing the agent to condition its policy on a task-relevant subgraph. The distribution for this mask is trained end-to-end alongside the agent's policy by maximizing an Evidence Lower Bound (ELBO) objective, which enables agents to jointly optimize their interaction topology and decision-making strategy using only local information.

Empirically, BayesG was evaluated on five adaptive traffic signal control benchmarks, including large-scale scenarios with up to 167 agents based on real-world New York City layouts. The results demonstrate that BayesG consistently and significantly outperforms a selection of MARL baselines comprising communication-based and non-communication-based methods.  BayesG achieves significantly higher coordination returns, faster convergence, and better scalability. Furthermore, case studies and ablation experiments validate the effectiveness of the learned graphs, showing they are interpretable and crucial for achieving efficient coordination by dynamically adapting to changing traffic conditions.

**Questions:**

1. What is the reasoning for selecting CommNet and NeuroComm as representative communication-based baselines for the experiments?

2. How does the proposed method perform compared to other methods that also learns a graph structure for communication, such as CASEC?

3. What is the probabilistic interpretation of using the usual policy gradient objective (also added with the entropy term) as the log likelihood function within the ELBO objective?

**Ethical Concerns:**

["NO or VERY MINOR ethics concerns only"]

**Final Justification:**

The authors addressed all the minor concerns I outlined except for my point regarding the method's novelty. Given the extensive work in similar problem settings, I believe it is crucial to develop a novel graph structure learning method to improve my scores. That being said, I am confident this is a borderline paper that is otherwise ready for acceptance.

**Limitations:**

Yes

**Paper Formatting Concerns:**

I did not find major formatting issues in the paper.

**Quality:**

3

**Strengths And Weaknesses:**

**Strength**

**(Major Strength - Quality - Experiment Design).** Aside from some small details that can be improved (see the weaknesses section below), the authors did a good job of designing and conducting experiments that demonstrate the paper's main claims. Aside from demonstrating the proposed methods' significantly higher returns compared to other methods, the ablation study and empirical analysis of the interpretability of the graph structure discovered by the method were strong aspects of the paper that I hope will remain in the final version.

**(Major Strength - Clarity).** Overall, the paper was quite well-written. It was easy to identify the motivation underlying the research, the main claims, and the intuition behind the method.

**(Major Strength - Clarity - Experiment Findings & Difference Between Compared Methods).** I believe the authors also did a great job highlighting the purpose of different experiments and what findings were discovered from each of them. At the same time, a sufficient description of the differences between the proposed method and other baseline methods/ablations was provided, making it easy for a reader to appreciate the significance of the comparisons made by the authors.

**(Minor Strength - Significance). Despite the proposed idea being relatively incremental, I believe the well-designed and well-executed experiments in this paper have demonstrated that the proposed method is a performant method, which, according to the experiments, also scales to settings with hundreds of agents. At the very least, I am confident that the MARL community, specifically those working on learning inter-agent communication to perform well in collaborative tasks, would view this as a competitive baseline for future work.**

**Weaknesses**

**(Minor Weakness - Originality - Incremental Method).** The idea of optimising the evidence lower bound (ELBO) to infer the appropriate graph structure for some GNN is something that has been explored in prior work [1,2]. At the same time, in the context of MARL, the idea of learning a local graph structure to focus on attending more relevant messages from a subset of neighbours has also been explored in prior work [3]. Since these prior work was not cited in the paper, I expect that the paper's final version will contain adequate references to these papers. That being said, the combination of these two ideas, along with the specific use of the usual RL objective as the log likelihood term in the ELBO evaluation, is something newly introduced in this paper.

**(Minor Weakness - Quality - Communication-based Baseline Selection).** It is also unclear to me why more recent communication-based methods, which also learn an appropriate graph structure for action selection, such as CASEC [3], are not selected as a baseline for this method. Furthermore, it is unclear why CommNet and NeuroComm, which are relatively older MARL methods, are instead selected as the representative baselines of the communication-based MARL methods.

**(Minor Weakness - Clarity - Likelihood Function Selection in ELBO).** I would also suggest providing stronger justification for the use of the usual policy gradient objective (i.e., the product between action log likelihood and returns added by the entropy term) as the log likelihood function. In particular, I believe the authors could draw a parallel between this log likelihood function and the probabilistic interpretations underlying soft policy iteration [4] (and more generally, maximum entropy reinforcement learning).

References:

[1] Kipf et al. 2018. Neural Relational Inference for Interacting Systems. ICML 2018.

[2] Löwe et al. 2022. Amortized Causal Discovery: Learning to Infer Causal Graphs from Time-Series Data. CLeaR 2022.

[3] Wang et al. 2022. Context-Aware Sparse Deep Coordination Graphs. ICLR 2022.

[4] Haarnoja et al. 2018. Soft Actor-Critic: Off-Policy Maximum Entropy Deep Reinforcement Learning with a Stochastic Actor. ICML 2018.

---

> ### Author Rebuttal · Authors · 2025-07-30
>
> >**On Baseline Selection and Comparison with Graph-Structured Methods like CASEC (Q1 & Q2)**
>
> We thank the reviewer for the thoughtful questions regarding our choice of baselines and the omission of recent graph-based methods like CASEC from our experimental comparisons.
>
> Our baseline selection is grounded in the theoretical setting of **Spatiotemporal Markov Decision Processes (Spatiotemporal-MDP)**, as formalized in Definition 1 of our paper. In Spatiotemporal-MDP, each agent’s transition dynamics and reward function are explicitly localized over its physically connected neighbors, meaning that the environment inherently imposes a structured graph topology (e.g., road networks in traffic signal control).
>
> Specifically, the local transition function `p_i(s_i' | s_{V_i}, u_i, u_{N_i})` depends only on the state-action tuples of agent $i$ and its neighbors $\mathcal{N}_i$, and the reward
> `R(s_{V_i}, u_{V_i})`
> is likewise defined over the local subgraph. The resulting learning setting is fully decentralized, grounded in real-world physical constraints, and lacks any centralized critic, global reward, or global coordination mechanism.
>
>
> In contrast, many recent graph-based MARL methods such as CASEC, DCG [1], DICG [2], SOP-CG [3], and GACG [4] are built on the Decentralized Partially Observable MDP (Dec-POMDP) framework. These methods operate under significantly different assumptions, including:
>
> - Access to or estimation of a **global state** $s \in S$,
>
> - A **shared global reward signal** $R(s, \mathbf{a})$ based on the joint action,
>
> - **Joint action-based transitions** $P(s^{\prime} \mid s, \mathbf{a})$,
>
> - And often, centralized training schemes with **global observability** or coordination.
>
> These assumptions make Dec-POMDP-based algorithms fundamentally incompatible with real-world physical systems governed by local dynamics, such as traffic networks, power grids, or sensor fields, where each agent only observes and affects its **local neighborhood**.
>
> Therefore, to ensure a fair and meaningful comparison aligned with our problem setting, we selected baselines that **respect the assumptions and structure of the Spatiotemporal-MDP framework**. Specifically, methods like CommNet, NeuroComm, and LToS are well-suited because:
>
> - They communicate **only with physically connected neighbors**.
>
> - They respect the **fixed topology** imposed by the environment graph.
>
> - They perform **local message passing** and **decentralized policy learning**, without relying on a global reward or state.
>
> - These baselines thus share the same structural and semantic assumptions as our proposed method.
>
> Finally, while CASEC and other Dec-POMDP-based methods represent important contributions to multi-agent coordination under different settings, we intentionally excluded them from our main comparisons due to their incompatibility with the physical and decentralized nature of ST-MDP. **We will add a clarifying discussion in the final version and cite these works in the related work section to clearly distinguish between the modeling assumptions and ensure transparency in our baseline choices.**
>
> References:
>
>  [1] Boehmer et al. Deep Coordination Graphs. ICML 2020.
>
>  [2] Liu et al. Deep Implicit Coordination Graphs. AAMAS 2021.
>
>  [3] Yan et al. Self-Organized Polynomial-Time Coordination Graphs. ICML 2022.
>
>  [4] Duan et al. Group-Aware Coordination Graph. IJCAI 2024.
>
> ---------------
> >**The probabilistic interpretation of using the usual policy gradient objective (Q3)**
>
> We thank the reviewer for this insightful question.
>
> In our framework, we adopt a probabilistic perspective on reinforcement learning. The trajectory data $D_i$  generated by each agent is treated as being sampled from an underlying distribution governed jointly by the agent's policy and the latent communication structure $Z_i$.
>
> **Why Use the Policy Objective as the Likelihood in the ELBO**
>
> In variational inference, the log-likelihood term `log p(D_i | Z_i, G_env_{V_i})` measures how well the latent structure $Z_i$  explains the data. We instantiate this term using the graph-conditioned actor loss $\mathcal{L}_{\theta,\varphi}$ , defined in Definition 4.
>
> Specifically:  $log \tilde{\pi}_i(a_i \mid \tilde{f}_i) \cdot \hat{A}_i$
> is treated as the data log-likelihood. This quantity provides task-aligned feedback on how suitable the sampled graph $Z_i$  is for policy optimization: **if a sampled structure leads to high advantage (good performance), it is assigned higher likelihood in the ELBO objective.** This ties the learning of latent graph structures directly to downstream task performance.
>
> This formulation follows the same probabilistic treatment as entropy-regularized RL, particularly the Soft Actor-Critic (SAC) framework [Haarnoja et al., 2018], which casts RL as probabilistic inference. In SAC, the optimal policy is derived by maximizing the expected log-likelihood of actions under a soft Q-function, plus an entropy bonus to ensure sufficient stochasticity and exploration. This regularization is critical for:
>
> - Improving exploration in high-dimensional action spaces.
>
> - Avoiding premature convergence to suboptimal deterministic policies.
>
> - Encouraging robustness to model misspecification and variance in gradients.
>
> Our actor loss mirrors this design, embedding stochasticity into policy optimization while preserving alignment with the Bayesian structure learning objective.
>
> **Why Include Mask Entropy $-\log q(Z_i; \phi_i)$**
>
> Analogous to policy entropy in SAC, we include a mask entropy term in our ELBO objective, defined over the variational distribution $q(Z_i; \phi_i)$ of edge masks. This term serves several key purposes:
>
> - Uncertainty Modeling: The entropy of the variational mask distribution captures uncertainty in the learned communication structure. In real-world settings with noisy and partially observable interactions, such uncertainty is inevitable and informative.
>
> - Regularization: Including mask entropy discourages premature collapse of the variational distribution to a deterministic mask. This is particularly important in early training when structural estimates are still evolving.
>
> - Diversity and Exploration: High entropy promotes diversity in sampled graphs, allowing agents to explore multiple plausible communication topologies. Over time, the posterior sharpens around those that consistently lead to higher policy performance.
>
> - Analytical Tractability: Since $q(Z_i)$ is a factorized Bernoulli distribution, its entropy can be computed in closed form using the sigmoid outputs $\sigma(\phi_{ij})$. This allows for efficient optimization.
>
> This design is aligned with the maximum entropy principle in SAC and variational inference theory, where entropy promotes robustness and avoids overfitting to noise or spurious correlations in structure learning.
>
> By treating the graph-conditioned policy objective as a proxy for the log-likelihood and incorporating entropy terms for both policy and structure, our ELBO formulation provides a principled probabilistic mechanism for joint optimization of communication and action strategies. We will clarify this probabilistic interpretation and its relation to Soft Actor-Critic in the final version.

---

> > ### Comment · Reviewer_2Jg6 · 2025-08-02
> > **Reply By Reviewer 2Jg6**
> >
> > Thank you for the detailed response to the points I raised.
> >
> > `We will add a clarifying discussion in the final version and cite these works in the related work section to clearly distinguish between the modeling assumptions and ensure transparency in our baseline choices.`
> >
> > This addresses my concern regarding the baseline selection.
> >
> > However, I have a follow-up suggestion regarding a more appropriate MARL objective under the *Spatiotemporal-MDP framework*. Currently, the outlined objective (Lines 97-98) appears to define the MARL objective only for one agent (i.e., agent $i$). Given that (unlike Dec-POMDP) the Spatiotemporal-MDP formulation seems to formalize a mixed-motive game where different agents may receive different reward scalars at every timestep, shouldn't the desired solution concept either be (i) achieving Nash Equilibrium or (ii) maximizing utilitarian welfare (i.e., sum of all agents' returns)? These objectives should better account for potentially undesirable outcomes when an agent selfishly maximizing its returns would lead to unwanted outcomes.
> >
> > `We will clarify this probabilistic interpretation and its relation to Soft Actor-Critic in the final version.`
> >
> > This also resolves my remaining concerns.
> >
> > Given that the rebuttal addresses my concerns and no new concerns arise from other reviewers' comments, I will keep my score as it is.

---

### Official Review · Reviewer_6Gbd · 2025-07-11

**Clarity:** 3
**Significance:** 3
**Originality:** 3
**Rating:** 4
**Confidence:** 4

**Summary:**

In this paper, Networked-MARL, decentralized agents struggle with local observability and fixed-graph communication, hindered by static methods or centralized demands. We propose a stochastic graph policy with sampled local subgraphs and introduce BayesG, a decentralized framework learning sparse interactions via Bayesian inference. BayesG excels in large-scale traffic control (167 agents), outperforming baselines in scalability and efficiency.

**Questions:**

Why not use direct communication but instead employ variational latent variables as the medium for information propagation? And could you explain the specific mechanisms behind why using variational latent variables as communication leads to better policy performance?

**Ethical Concerns:**

["NO or VERY MINOR ethics concerns only"]

**Quality:**

3

**Strengths And Weaknesses:**

Strengths
Using a variational approach with latent variables as carriers for information propagation.

---

> ### Author Rebuttal · Authors · 2025-07-29
>
> >**Why not use direct communication? Why use variational latent variables for message masking?**
>
> We appreciate this insightful question. There are two layers to unpack here:
>
> (1) Why not direct communication over the full environment graph?
>
>  While the environment-provided physical graph (e.g., road topology in traffic networks) provides a grounded prior for interactions, communicating over the entire neighborhood indiscriminately can lead to redundant or noisy information aggregation. Our ablation in Section 5.4 (Figure 5) compares no masking (i.e., direct message passing on the environment graph) with BayesG’s learned mask, and shows that direct communication performs consistently worse. Therefore, selectively filtering communication is essential to avoid distraction and enhance relevance.
>
> (2) Why use a variational latent variable instead of learning a deterministic mask?
>
>  Rather than directly learning a fixed mask, we model the mask as a stochastic latent variable sampled from a variational distribution $q(Z_i; \phi)$. This design brings several key advantages:
>
>   - Captures uncertainty in graph structure, which is particularly important under partial observability or sparse feedback in MARL.
>
>   - Encourages exploration in structural space, enabling agents to discover better interaction topologies during training.
>
>  - Incorporates structured priors, such as a Bernoulli sparsity prior, to promote minimal and efficient graphs.
>
> - Supports entropy-based regularization, which smooths optimization and improves training stability.
>
> In summary, direct communication is ineffective due to a lack of relevance filtering; deterministic masking fails to model structural uncertainty. By contrast, our variational latent mask provides a principled, adaptive, and uncertainty-aware mechanism that leads to improved decentralized decision-making, as validated by ablation results.
>
>  >**Mechanism for improved policy performance and empirical evidence (Section 5.4)**
>
>  The learning signal for $Z_t$ is provided via the graph-conditioned actor loss $\mathcal{L}_{\theta,\phi}$  (see Definition 4), which appears in the ELBO's likelihood term. This loss connects interaction topology learning to downstream task performance: **if a sampled graph leads to better policy outcomes, it receives a higher likelihood.** This tight coupling ensures that the learned graphs are task-driven and performance-aligned.
>
> In our ablation study on different graph masking strategies (Section 5.4), we compare:
>
> - No masking (i.e., using the full environment graph),
>
> - Random masking, and
>
> - Our learned Bayesian mask (BayesG).
>
>  The learned mask consistently outperforms the baselines, showing that task-aware variational latent variables yield more effective communication graphs, ultimately leading to better decentralized policy performance.

---

### Public Comment · ~Wei_Duan2 · 2026-02-13
**Correction to ELBO Derivation (Equations 33 and Final Objective)**

**Correction to ELBO Derivation (Equations 33 and Final Objective)**

We would like to clarify and correct an inconsistency in our original submission regarding the derivation of the ELBO objective.

**Issue Identified:**

In the original version, there was an error in how the prior term from Equation 33 was combined with the entropy term in the Final Objective. Specifically:

**Equation 33 (Prior term - CORRECT):**

$$E_q[\\log p(Z_i)] = \\sum_{j \\in N_i}[\\sigma(\\phi_{ij}) \\log \\lambda + (1-\\sigma(\\phi_{ij}))\\log(1-\\lambda)]$$

**Original Final Objective (INCORRECT):** The combination incorrectly appeared as:

$$\\sum_{j \\in N_i}[\\lambda \\log \\sigma(\\phi_{ij}) + (1-\\lambda)\\log(1-\\sigma(\\phi_{ij}))]$$

This improperly swapped λ and σ(φᵢⱼ) in the logarithm arguments.

**Correction:**

The correct ELBO should be:

$$L_{\\text{ELBO}} = E_q[-L_{\\theta,\\varphi}] - \\sum_{j \\in N_i} \\text{KL}(q(Z_{ij};\\phi_{ij}) \\| p(Z_{ij}))$$

where $q(Z_{ij};\\phi_{ij}) = \\text{Bernoulli}(\\sigma(\\phi_{ij}))$ and $p(Z_{ij}) = \\text{Bernoulli}(\\lambda)$.


**Impact:**

This correction does not affect the qualitative conclusions of our work or the comparative experimental results. While the implementation utilized the separate-term formulation which mirrored the swap in the original manuscript, the resulting objective still functions as a valid regularizer pulling the variational distribution toward the prior. We have verified that the optimization landscape remains consistent and the model converges to the intended sparse solutions. We have updated the mathematical derivation in our latest manuscript on arXiv (https://arxiv.org/abs/2509.16606) to ensure formal correctness and apologize for any confusion.

We thank the reviewers/readers for their careful attention to mathematical detail.

---

### Note · Authors · 2025-08-14

We thank the reviewers and area chair for the thoughtful and constructive feedback. Below, we briefly emphasize the key contributions and clarifications from our rebuttal that we believe are crucial for the decision process.

**1. Probabilistic Graph Masking via ELBO**

Our use of a variational latent mask (rather than direct communication or deterministic masking) is both principled and impactful. By framing the communication mask as a stochastic latent variable and optimizing it via an ELBO, our method brings multiple advantages:

- It models uncertainty over graph structure, crucial for MARL under partial observability and noisy feedback.
- It encourages structural exploration and discovers efficient, task-adaptive topologies.
- It supports entropy-based regularization, improving optimization and training stability.

Our probabilistic view, including treating the actor loss as a log-likelihood term in the ELBO (akin to Soft Actor-Critic), provides a cohesive framework for joint structure–policy learning.

**2. Suitability of Baselines and Setting**

We focus on **decentralized, spatiotemporal settings with local observability**—distinct from the Dec-POMDP assumption of works like CASEC, DCG, or SOP-CG. These works rely on centralized training, global reward/state access, or joint action modeling—assumptions incompatible with decentralized physical systems like traffic or sensor networks. We intentionally select baselines (e.g., CommNet, LToS) that conform to our decentralized, physically grounded setup. We will further clarify this distinction and cite relevant works in the revision.

**3. Relation to Prior Sampling-based Methods**

We appreciate the pointer to related works on link sampling (e.g., Lin et al., Anand & Qu). Our method differs in both object and purpose: instead of fixed or uniformly sampled active links, we learn **a per-agent, per-edge posterior over 1-hop neighbors** conditioned on state/trajectory, jointly trained with the policy. This yields communication-level adaptivity with stronger task alignment. While those works provide valuable theoretical tools (e.g., µ-decay, mean-field approximations), we view them as complementary and plan to explore extensions of their analyses to our setting.

---

### Decision · Program_Chairs · 2025-09-17

**Decision:**

Accept (poster)

**Comment:**

This paper presents BayesG, a decentralized actor–critic approach for networked multi-agent reinforcement learning (MARL), that learns sparse, context-aware communication graphs in spatiotemporal MDPs using Bayesian variational inference. The paper is mostly well-written and has strong performance in the traffic signal control tasks.

There are some concerns about the novelty of the approach and the presentation, but the approach is interesting and performs well. As noted in the reviews, the paper should be updated to more clearly discuss the assumptions needed, motivation for algorithmic choices, relationship with other methods, and to fix typos and other grammatical issues.

The author response and discussion was helpful in addressing these points but the paper should be updated further to address these issues.